# An open cortico-basal ganglia loop allows limbic control over motor output via the nigrothalamic pathway

Sho Aoki[1,2,3,4†]*, Jared B Smith[1†], Hao Li[1†], Xunyi Yan[1], Masakazu Igarashi[2,4], Patrice Coulon[5], Jeffery R Wickens[2], Tom JH Ruigrok[3]*, Xin Jin[1]*

[1]Molecular Neurobiology Laboratory, Salk Institute for Biological Studies, La Jolla, United States; [2]Neurobiology Research Unit, Okinawa Institute of Science and Technology, Okinawa, Japan; [3]Department of Neuroscience, Erasmus Medical Center Rotterdam, Rotterdam, Netherlands; [4]Japan Society for the Promotion of Sciences, Tokyo, Japan; [5]Institut des Neurosciences de la Timone, Centre National de la Recherche Scientifique (CNRS), Aix-Marseille Université, Marseille, France

**Abstract** Cortico-basal ganglia-thalamocortical loops are largely conceived as parallel circuits that process limbic, associative, and sensorimotor information separately. Whether and how these functionally distinct loops interact remains unclear. Combining genetic and viral approaches, we systemically mapped the limbic and motor cortico-basal ganglia-thalamocortical loops in rodents. Despite largely closed loops within each functional domain, we discovered a unidirectional influence of the limbic over the motor loop via ventral striatum-substantia nigra (SNr)-motor thalamus circuitry. Slice electrophysiology verifies that the projection from ventral striatum functionally inhibits nigro-thalamic SNr neurons. In vivo optogenetic stimulation of ventral or dorsolateral striatum to SNr pathway modulates activity in medial prefrontal cortex (mPFC) and motor cortex (M1), respectively. However, whereas the dorsolateral striatum-SNr pathway exerts little impact on mPFC, activation of the ventral striatum-SNr pathway effectively alters M1 activity. These results demonstrate an open cortico-basal ganglia loop whereby limbic information could modulate motor output through ventral striatum control of M1.
DOI: https://doi.org/10.7554/eLife.49995.001

*For correspondence:
saoki@salk.edu (SA);
t.ruigrok@erasmusmc.nl (TJHR);
xjin@salk.edu (XJ)

†These authors contributed equally to this work

**Competing interests:** The authors declare that no competing interests exist.

## Introduction

Cortico-basal ganglia circuits are crucial for emotional, cognitive, and sensorimotor functions in health and disease (*Doya, 2000*; *Floresco, 2015*; *Gerdeman et al., 2003*; *Graybiel et al., 1994*; *Gunaydin and Kreitzer, 2016*; *Hikosaka et al., 2000*; *Jahanshahi et al., 2015*; *Jin and Costa, 2015*; *Marchand, 2010*; *Vaghi et al., 2017*; *Yin and Knowlton, 2006*). Virtually all cortical regions project to the striatum, the main input nucleus of the basal ganglia, which plays an important role in guiding behavior (*Hintiryan et al., 2016*; *Hooks et al., 2018*; *Voorn et al., 2004*; *Witten et al., 2010*; *Yin et al., 2009*; *Znamenskiy and Zador, 2013*). For example, the 'sensorimotor' dorsolateral striatum (DLS), which receives inputs from motor cortex, plays a role in executing body movements (*Barbera et al., 2016*; *Rueda-Orozco and Robbe, 2015*; *Yin, 2010*). However, the 'limbic' ventral striatum (VS), which receives limbic but not motor cortical input, also alters behavioral output including locomotion activity, approach/avoidance behaviors, and recovery of skilled movement after spinal cord injury (*Britt et al., 2012*; *Floresco, 2015*; *Saunders and Robinson, 2012*; *Sawada et al., 2015*). These findings suggest that for a unified behavioral output, information across the different modalities must be integrated into motor circuits to drive action appropriately (*Mogenson et al., 1980*). Though some studies have implicated mechanisms for limbic-motor interactions in the

dopaminergic system (*Beier et al., 2015*; *Belin and Everitt, 2008*; *Haber et al., 2000*; *Lerner et al., 2015*; *Watabe-Uchida et al., 2012*; *Yang et al., 2018*), how limbic information ultimately reaches motor circuitry remains largely unknown, as do the characteristics of its influence.

Cortico-basal ganglia-thalamocortical loops have been largely conceptualized as closed, functionally segregated loops, in which limbic, associative, and sensorimotor information are processed in parallel (*Alexander et al., 1986*; *Deniau et al., 1996*; *Haber, 2003*; *Kim and Hikosaka, 2015*; *Montaron et al., 1996*; *Parent and Hazrati, 1995*). Alternatively, older studies proposed a 'funnel-like' architecture for basal ganglia output, such that each loop provides some input to the motor circuit (*Allen and Tsukahara, 1974*; *Kemp and Powell, 1971*). A 'partially-open' loop architecture in the cortico-basal ganglia circuitry has been suggested from primate studies (*Joel and Weiner, 1994*; *Kelly and Strick, 2004*; *Miyachi et al., 2006*), but this previous evidence is incomplete and the precise anatomical basis underlying connections between functionally distinct loops has not been identified. This lack of clarity in the cortico-basal ganglia connections is due to technical limitations, which include the lack of sophisticated viral tools and the complicated geometry of basal ganglia nuclei. As a result, studies have largely focused on mapping monosynaptic inputs from cortex to the striatum (*Hintiryan et al., 2016*; *Hooks et al., 2018*; *Voorn et al., 2004*), emphasizing the topographic organization at the level of cortico-striatal projections. To date, it is incompletely understood how these distinct 'channels' proceed through the rest of basal ganglia-thalamo-cortical circuitry and whether they interact at all.

In the current study, we focused on limbic and motor cortico-basal ganglia loops that originate from the medial prefrontal cortex (mPFC) and primary motor cortex (M1). These limbic and motor loops provide a model to investigate how distinct cortico-basal ganglia loops are organized throughout basal ganglia output, which we mapped using multiple genetic and viral tracing tools. In addition to the closed loop within each functional domain, we identified a novel one-way interaction across these cortico-basal ganglia loops, in which the limbic loop exerts a unidirectional influence over the motor loop that is mediated by ventral striatum-medial SNr-motor thalamus circuitry. Using slice physiology and optogenetics, we show that VS functionally inhibits medial SNr neurons that project to motor thalamus. We then characterized the influence of limbic and motor striato-nigral outputs onto downstream cortical targets by optogenetically activating striato-nigral terminals from VS or DLS and by recording neuronal activity in mPFC and M1. Consistent with our anatomical findings, the in vivo recording experiments showed, in addition to the within-loop activation in which VS activated mPFC and DLS activated M1, significant activation of M1 when VS terminals in the SNr were stimulated. Conversely, DLS output did not effectively modulate activity in mPFC. Together, these results demonstrate an open cortico-basal ganglia-thalamocortical loop through which VS can modulate M1 activity, providing new insights into the limbic control over motor output in health and disease.

## Results

### Trans-synaptic tracing using wild-type rabies virus reveals both closed and open cortico-basal ganglia-thalamocortical loops

To visualize the cortico-basal ganglia loops, we first mapped the input-output connections of the rodent striatum with the primary motor cortex (M1), secondary motor cortex (M2), and medial prefrontal cortex (mPFC) by injecting a mixture of cholera toxin b subunit (CTb, non-trans-synaptic, bi-directional tracer) and a retrogradely transported poly-synaptic, wild-type rabies virus (Wt-RABV) into each cortical area in rats (see Materials and methods). This strategy allowed us to compare the topography of the cortico-striatal input with that of striatal output neurons that multi-synaptically connect to the same area of cortex via the canonical basal ganglia direct pathway (i.e. via striato-nigro-thalamo-cortical circuitry) (*Figure 1A*). Wt-RABV has been repeatedly validated as a means of trans-synaptically tracing circuits retrogradely, in a survival-time-dependent manner (*Aoki et al., 2019*; *Kelly and Strick, 2004*; *Suzuki et al., 2012*; *Ugolini, 2010*). Prior studies have established that 66–70 hr is adequate survival time for 3rd-order infection of Wt-RABV without 4th-order infection in rats (*Aoki et al., 2019*; *Suzuki et al., 2012*), so this procedure could determine tri-synaptic connections originating from striatum to cortex via the direct pathway (*Kelly and Strick, 2004*; *Miyachi et al., 2006*).

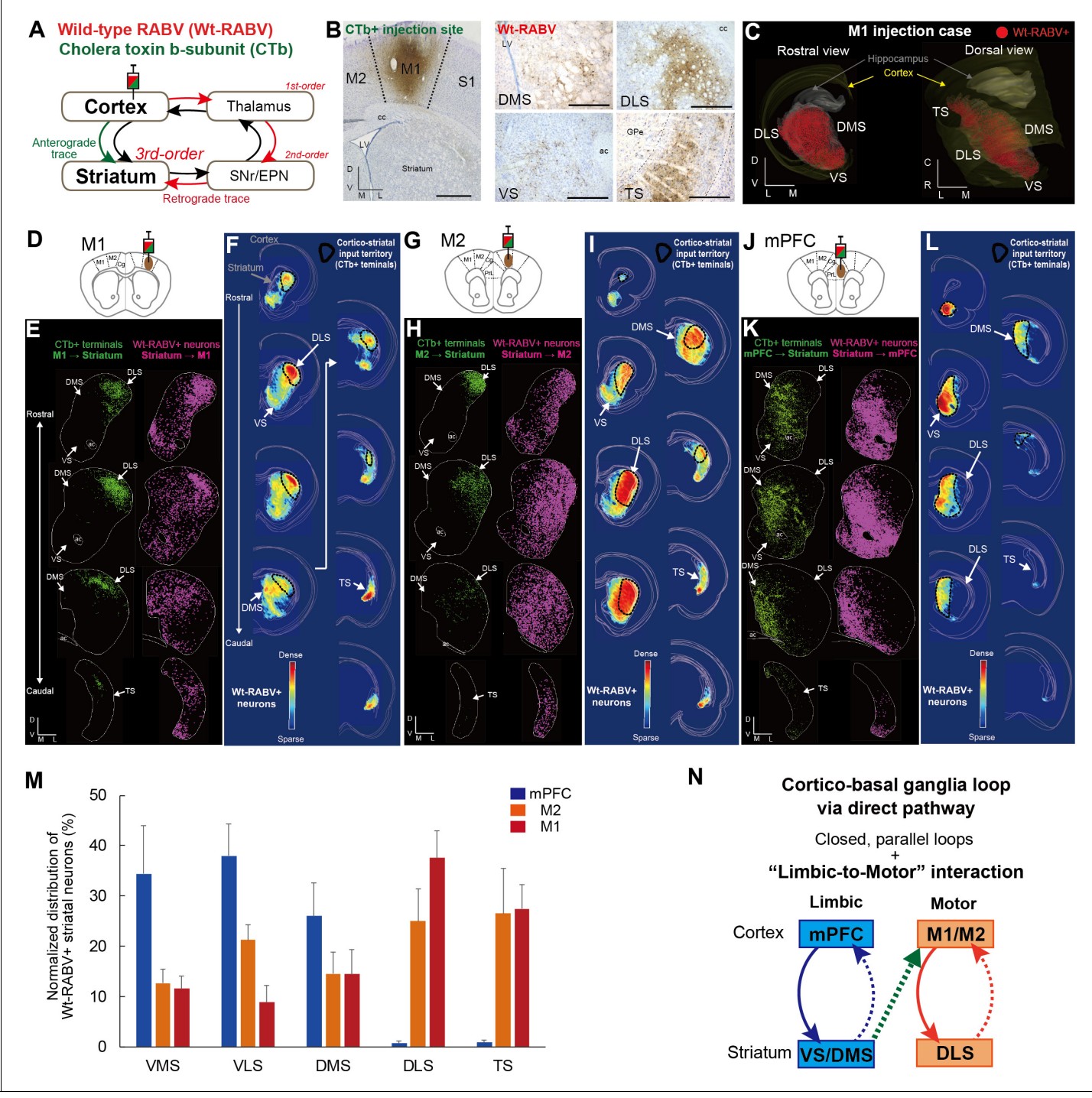

**Figure 1.** Trans-synaptic wild-type rabies tracing reveals both closed and open cortico-basal ganglia loops. (**A**) The strategy to label striatal neurons connecting to the cerebral cortex by Wt-RABV trans-synaptic retrograde tracing, and CTb-based non-trans-synaptic anterograde tracing for mapping cortico-striatal terminals. (**B**) Example image of Wt-RABV/CTb injection into M1 (left). After 66–70 hr of survival time, Wt-RABV was transfected up to 3rd-order neurons, which were found in various striatal subregions (right). Scale bars, 1 mm (left), 500 μm (right). (**C**) 3D-reconstruction of Wt-RABV+ striatal neurons from the M1 injection case shown in (**B**). The two different angles emphasize the presence of Wt-RABV+ neurons throughout all of the striatum (VS, DMS, DLS, and TS). (**D**) Schema of Wt-RABV/CTb injection in M1. (**E**) Anterogradely labeled CTb+ cortico-striatal terminals (green) and retrogradely labeled Wt-RABV+ striatal neurons (purple) from the M1 injection case shown in (**B**). (**F**) Density map showing the distribution of Wt-RABV+ neurons throughout the striatum from M1 injection. Black contours indicate approximate areas receiving cortico-striatal inputs from M1. Color maps indicate the intensity of Wt-RABV+ labeling. (**G–I**) The same analyses for Wt-RABV/CTb injection in M2. (**J–L**) The same analyses for Wt-RABV/CTb injection in mPFC. (**M**) Normalized distribution of Wt-RABV+ neurons across five striatal regions (VMS, VLS, DMS, DLS, TS) showing differences

*Figure 1 continued on next page*

*Figure 1 continued*

depending on cortical injection sites (mPFC, n = 3; M2, n = 4; M1, n = 4). Data are expressed as mean ± SEM. Two-way ANOVA, Interaction (Injection site x Labeled striatal regions): $F_{(8,40)}$ = 8.208, p<0.0001. (N) Summary diagram indicates the closed limbic and motor loops, as well as the unidirectional limbic-to-motor interaction. Monosynaptic and multi-synaptic pathways are shown as solid and dashed lines, respectively. Abbreviations: cc, corpus callosum; ac, anterior commissure; LV, lateral ventricle.

DOI: https://doi.org/10.7554/eLife.49995.002

The following figure supplements are available for figure 1:

**Figure supplement 1.** Comparison of Wt-RABV labeling patterns between cases with shorter (58 hr) and longer (70 hr) survival time.
DOI: https://doi.org/10.7554/eLife.49995.003
**Figure supplement 2.** Detailed procedure for the anatomical analysis of Wt-RABV/CTb tracing.
DOI: https://doi.org/10.7554/eLife.49995.004
**Figure supplement 3.** Comparisons of the topography of cortico-striatal inputs and striatal outputs to the cortex.
DOI: https://doi.org/10.7554/eLife.49995.005

In addition to the classical basal ganglia direct pathway output to cortex, previous studies have identified direct projections from external globus pallidus (GPe) and subthalamic nucleus (STN) to the frontal cortex (*Chen et al., 2015*; *Jackson and Crossman, 1981*; *Saunders et al., 2015*). To account for these potential confounds, we investigated the alternative routes for mediating Wt-RABV transfection by comparing patterns of Wt-RABV+ labeling between shorter (58 hr) and longer (70 hr) survival times after Wt-RABV injections in M1 (*Figure 1—figure supplement 1*). The 58 hr survival time resulted in dense labeling of Wt-RABV+ cells in motor thalamus, as well as obvious Wt-RABV+ cells in SNr and thalamic reticular nucleus (TRN), indicating that the motor thalamus is the site of 1st order labeling, followed by SNr and TRN as 2nd order neurons (*Figure 1—figure supplement 1B and C*). By contrast, within this 58 hr survival time (which includes both 1st and 2nd order neurons), there was almost no labeling in GPe, STN and striatum, although each of these regions showed intense Wt-RABV+ labeling with 70 hr of survival time (*Figure 1—figure supplement 1B and C*). Therefore, Wt-RABV+ labeling in GPe, STN and striatum should be considered primarily 3rd order neurons, suggesting that GPe and STN inputs to cortex cannot predominantly account for the striatal labeling observed after 70 hr of survival time. This finding allows us to infer that the striatal labeling in the following experiments is largely mediated by the canonical striato-nigro-thalamo-cortical pathway (*Figure 1A*), although we cannot completely rule out the possible minor contribution of direct GPe and STN inputs to cortex with this methodology.

A local injection of Wt-RABV/CTb in M1 resulted in anterograde CTb+ and retrograde Wt-RABV+ labeling in the striatum (*Figure 1B and C*, *Figure 1—figure supplement 2A and I*). We first mapped both CTb+ terminals and Wt-RABV+ striatal neurons and compared their territories within the striatum (*Figure 1D–1F*). The M1 injection revealed CTb+ terminals that overlapped with Wt-RABV+ neurons in the dorsolateral striatum (DLS), which is known to process motor information (*Hintiryan et al., 2016*; *Voorn et al., 2004*), thus providing evidence for a closed motor loop between M1 and DLS. However, M1 injections also labeled a large number of Wt-RABV+ neurons outside the CTb-labeled zone in DLS, including the dorsomedial (DMS), ventral (VS), and tail of striatum (TS) (*Figure 1E and F*, and *Figure 1—figure supplement 2I*), indicating an open loop in addition to the closed loop. This finding is consistent with the VS output to M1 in primate studies (*Kelly and Strick, 2004*; *Miyachi et al., 2006*). These data suggest that striatal subregions associated with emotional and cognitive (VS and DMS) functions, as well as sensory processing (TS), can target M1 via basal ganglia output, despite having no direct inputs from M1. A similar pattern of labeling was observed following M2 injections, where VS, DMS and TS can target M2 without receiving its direct inputs (*Figure 1G–1I*, *Figure 1—figure supplement 2C, D and J*), confirming an open loop in addition to the closed motor loop. In marked contrast, Wt-RABV/CTb injections into mPFC (including prelimbic (PrL) and cingulate cortex) revealed a closed limbic loop, where CTb+ terminals and Wt-RABV+ neurons, which largely overlapped, were found only in DMS and VS, with no labeling observed in DLS or TS (*Figure 1J–1L*, *Figure 1—figure supplement 2B, E, F and K*). We quantified the Wt-RABV+ cells across five striatal regions and found that the distributions were significantly different based on cortical injection site (*Figure 1M*, main effect of interaction, $F_{(8,40)}$ = 8.208, p<0.0001), suggesting a largely closed cortico-basal ganglia loop architecture within both the limbic and motor domains. However, while the absence of Wt-RABV+ cells in DLS following

mPFC injections indicates that the motor loop does not affect the limbic loop through the direct pathway, the limbic loop can influence motor circuits as revealed by the significant presence of VS labeling from M1/M2 injections (*Figure 1—figure supplement 2G*, $t_9$ = 4.229, p=0.0022). These findings confirm the existence of closed-loops within both limbic and motor domains in the cortico-basal ganglia circuitry, but importantly, they also reveal that limbic information originating from ventral striatum can also influence motor cortex via the basal ganglia direct pathway output (*Figure 1N*).

We also compared the distribution of cortico-striatal terminals and striatal output neurons connecting to the cortex (*Figure 1—figure supplement 3*). After measuring the intensity of cortico-striatal projections from mPFC or M1 on the basis of luminance analysis of fluorescence, we estimated the distribution of their axonal terminals across five striatal regions (VMS, VLS, DMS, DLS, and TS; *Figure 1—figure supplement 3A*). As reported earlier (*Hintiryan et al., 2016*; *Voorn et al., 2004*), M1 projects to DLS almost exclusively, whereas mPFC innervates mainly DMS and to a lesser degree VLS and VMS (*Figure 1—figure supplement 3A*). A comparison between mPFC inputs to the striatum and its outputs to the mPFC showed a significant interaction ($F_{(4,15)}$ = 8.022, p=0.0011). In particular, VMS had a mismatch in which it receives a few inputs but has many outputs (*Figure 1—figure supplement 3C*), suggesting the possibility that VMS provides a common source for the open loop structure (*Miyachi et al., 2006*). There was also significant interaction between M1 striatal inputs and striatal outputs to the M1 ($F_{(4,20)}$ = 15.5, p<0.0001; *Figure 1—figure supplement 3D*). This result indicates that only DLS receives M1 inputs but that multiple striatal regions (VMS, VLS, DMS, DLS and TS) connect to M1, implying that all striatal outputs are funneled into M1 via basal ganglia direct pathway output.

## Monosynaptic modified rabies tracing confirms the limbic-to-motor connectivity via the striato-nigro-thalamic pathway

To verify that the pattern of striatal Wt-RABV labeling from motor cortex injections is mediated by the nigro-thalamic route, we employed a novel viral strategy to identify striatal outputs connecting to thalamic motor nuclei, namely ventro-anterior and ventro-lateral (VA-VL) thalamus (*Figure 2A*). To this end, a mixture of AAVretro.Cre and AAV.FLEX.tdTomato was injected into VA-VL. This strategy had two consequences. First, recombination of Cre and FLEX.tdTomato defined the injection site in VA-VL (*Figure 2B*) and revealed their thalamo-cortical terminals in M1 and M2, but not in mPFC (*Figure 2C*). Note that a small fraction of tdTomato+ cells was found in latero-dorsal thalamus (LD, *Figure 2B*), but this nucleus does not receive SNr projections (Figure 4) and thus it should not affect the interpretation of these experiments. Second, the use of the AAVretro serotype induced retrograde Cre expression in SNr neurons projecting to VA-VL (*Tervo et al., 2016*). Concomitant injections of Cre-dependent AAV rabies-helper viruses (TVA and RG) and subsequent injection of GFP-expressing EnvA-pseudotyped G-deleted rabies virus (Rt-RABV-GFP) into SNr enabled subcircuit-specific rabies-based retrograde tracing, specifically from SNr neurons projecting to motor thalamus via the basal ganglia direct pathway output (*Wickersham et al., 2007*). Analysis of midbrain nuclei revealed mCherry+ cells in SNr, with virtually no co-localization with tyrosine hydroxylase (TH)-positive cells (less than 4%), indicating that only non-dopaminergic SNr neurons project to VA-VL (*Figure 2D*, *Figure 2—figure supplement 1*). Starter cells defined by mCherry+ and Rt-RABV-GFP+ (*Callaway and Luo, 2015*) were located throughout the medial to lateral extent of SNr (*Figure 2D and E*). A further control experiment of the Rt-RABV tracing without the injection of rabies glycoprotein was conducted to alternatively determine the restriction of starter cells in SNr (*Figure 2—figure supplement 1F–H*). Transfection of the monosynaptic rabies virus was observed throughout the striatum, including VS, DMS, DLS and TS (*Figure 2F–2G*). This result is consistent with the wild-type rabies tracing from M1, and they shared almost identical quantitative distributions (*Figure 2H*). These findings support our Wt-RABV data demonstrating that all striatal regions, including limbic VS, reach motor cortex through the basal ganglia direct pathway, that is, by the striato-nigro-thalamic route (*Figure 2I*).

As a downstream target of ventral striatum (*Heimer et al., 1982*; *Smith et al., 2009*), we analyzed Wt-RABV and Rt-RABV labeling in ventral pallidum (VP) to address whether limbic-to-motor connectivity exists through this nucleus. Labeling in VP was present following Wt-RABV injections into mPFC, but not after Wt-RABV tracing from M1 (*Figure 2—figure supplement 2A–B*). There was almost no labeling in VP in Rt-RABV tracing from SNr neurons projecting to VA-VL, as opposed

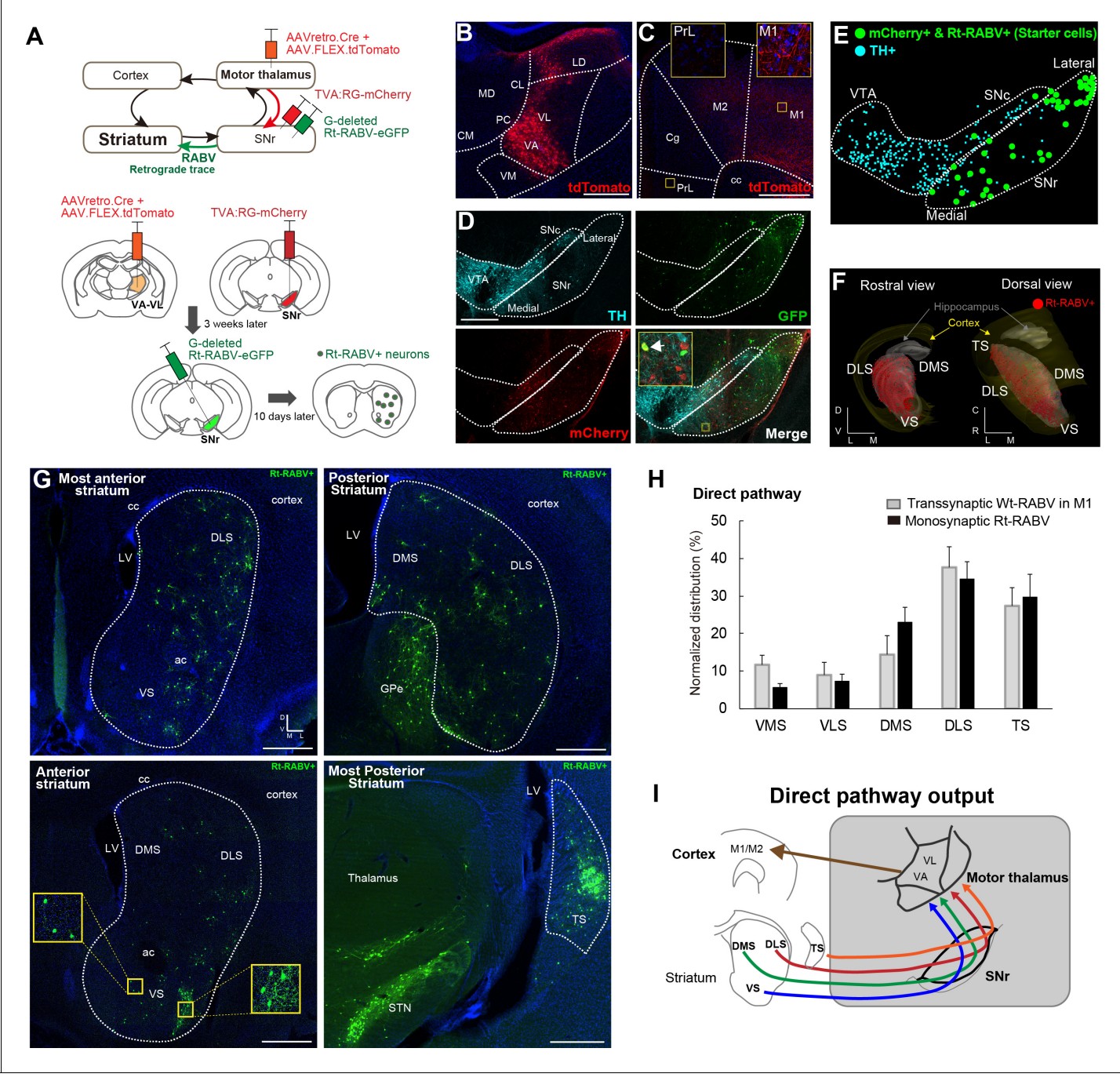

**Figure 2.** Subcircuit-specific modified rabies tracing confirms limbic-to-motor connectivity through the direct pathway. (**A**) Strategy to identify striatal neurons projecting to the specific SNr subpopulation that projects to VA-VL motor thalamus. (**B, C**) Images of the injection site in VA-VL thalamus (**B**) and their thalamo-cortical terminals in M1, but not in mPFC (**C**). Note that there are tdTomato+ cells in LD, but this part of thalamus does not receive SNr inputs (*Figure 4*). Scale bar, 500 μm. (**D**) Image of TVA-mcherry+ and Rt-RABV(GFP)+ cells in SNr. Immunohistochemistry for tyrosine hydroxylase (TH) revealed almost no co-localization of TH+ cells with mCherry+ SNr neurons projecting to VA-VL (<4%, see *Figure 2—figure supplement 1*). An arrow in the inset indicates a representative example of a starter cell with mCherry+ and Rt-RABV-GFP+. Scale bar, 200 μm. (**E**) Digital reconstruction of starter cells that are defined as mCherry+ and Rt-RABV-GFP+ neurons (green), relative to TH+ dopamine neurons (blue). (**F**) 3D-reconstruction of Rt-RABV+ striatal neurons. (**G**) Rt-RABV+ neurons at different anterior-posterior levels of striatum, which are found in VS, DMS, DLS and TS. Note that densely labeled GFP+ cells in TS are not starter cells as there are no TVA.mCherry+ cells. Scale bar, 500 μm. (**H**) Normalized distribution of Rt-RABV+ striatal cells (n = 6), compared to Wt-RABV tracing from M1 (n = 4) across five striatal regions. Data are expressed as mean ± SEM. Two-way ANOVA showing no significant effect of interaction (Injection site x Labeled striatal regions): $F_{(4,40)}$ = 0.8722, p=0.4891. (**I**) Summary diagram showing

*Figure 2 continued on next page*

*Figure 2 continued*

that through the direct pathway, VS, DMS, DLS and TS connect to motor thalamus, which in turn projects to motor cortex. Abbreviations: MD, mediodorsal; CM, centromedial; PC, paracentral; CL, centrolateral: LD, lateral dorsal thalamus; PrL, prelimbic cortex; Cg, cingulate cortex.

DOI: https://doi.org/10.7554/eLife.49995.006

The following figure supplements are available for figure 2:

**Figure supplement 1.** Detailed analysis for the starter cell population in recombinant rabies tracing of nigro-motor thalamic cells.

DOI: https://doi.org/10.7554/eLife.49995.007

**Figure supplement 2.** The absence of polysynaptic connections from ventral pallidum to motor thalamus and to motor cortex.

DOI: https://doi.org/10.7554/eLife.49995.008

to the dense labeling in GPe and STN (*Figure 2—figure supplement 2C and D*). This result suggests that the limbic-to-motor interaction does not occur through the ventral striatum to ventral pallidum pathway (*Figure 2—figure supplement 2E*).

## Both the medial and lateral SNr innervate motor thalamus

Since virtually all striatal regions have outputs to motor thalamus (*Figure 2*), it follows that there must be convergence from the entire SNr onto thalamo-cortical neurons projecting to M1. To test this hypothesis, we next injected AAVretro.Cre into M1, with TVA.RG and Rt-RABV-GFP in motor thalamus, enabling us to identify the SNr synaptic inputs to thalamocortical neurons that innervate M1 specifically (*Figure 3A*). Starter cells defined as both TVA.mCherry+ and Rt-RABV-GFP+ were found in VA-VL, whereas the adjacent thalamic reticular nucleus (TRN) showed only Rt-RABV-GFP+ cells (*Figure 3B*), suggesting that the primary starter cells are located selectively in VA-VL, and that labeling in TRN resulted from trans-synaptic rabies transfection. To further validate that our tracing was specific to thalamo-M1 cells, we analyzed labeling in cortex and cerebellar output nuclei, both of which are known to project to motor thalamus (*Aumann et al., 1994*; *Bostan et al., 2013*; *Hooks et al., 2013*; *Hoover and Strick, 1999*; *Kelly and Strick, 2003*; *Terashima et al., 1987*; *Yamawaki and Shepherd, 2015*). Cortico-thalamic GFP+ cells were found exclusively in motor cortex, but not in cingulate or prelimbic cortices (*Figure 3C*), and GFP+ cerebello-thalamic cells were located in the dentate and interpositus nuclei (*Figure 3D*), demonstrating the specificity of our tracing to motor thalamus. Most importantly, we found trans-synaptically labeled Rt-RABV-GFP+ cells in both medial and lateral SNr, covering the entire territory of the region (*Figure 3E and F*, *Figure 3—figure supplement 1A and B*). These findings provide strong evidence for the convergence of limbic, associative, and sensorimotor information from both medial and lateral SNr onto thalamo-M1 cells, and suggest that all basal ganglia outputs provide some amount of convergence into a 'funnel' towards the motor cortex (*Figure 3G*).

## Limbic output merges with motor circuits through ventral striatum – medial SNr – motor thalamic projections

To identify where the limbic output converges onto motor circuits via the direct pathway, we systemically mapped the topography of each synaptic step through the striato-nigro-thalamo-cortical pathway (*Figure 4* and *Figure 4—figure supplement 1*). Paired Cre-dependent AAV injections of GFP and tdTomato were made into DS and VS as well as DLS and DMS in D1-Cre mice, which produced segregated terminal fields in SNr (*Figure 4A–4F*, *Figure 4—figure supplement 1A–C*), consistent with the topography observed in a previous study (*Deniau et al., 1996*). In addition, DMS and DLS were found to project topographically to the EPN (*Figure 4C and F*), whereas VS innervates the adjacent lateral hypothalamus (LH) (*Figure 4C*), indicating that this alternative direct pathway output nucleus, EPN, is not how limbic information reaches motor cortex. Importantly, DLS, DMS and VS terminated in different SNr subregions, with VS specifically innervating the most medial region of SNr (*Figure 4B*). These data suggest that limbic and motor information remain largely segregated at this stage of basal ganglia output.

We next compared the projections from medial-SNr and lateral-SNr to thalamus by injecting the same Cre-dependent AAV-GFP or –tdTomato in the SNr of parvalbumin (PV)-Cre mice (*Figure 4G–4I* and *Figure 4—figure supplement 1D*), as >80% of SNr GABA neurons are PV-positive (*González-lez-Hernández and Rodríguez, 2000*; *Lee and Tepper, 2007*). SNr targeted multiple thalamic

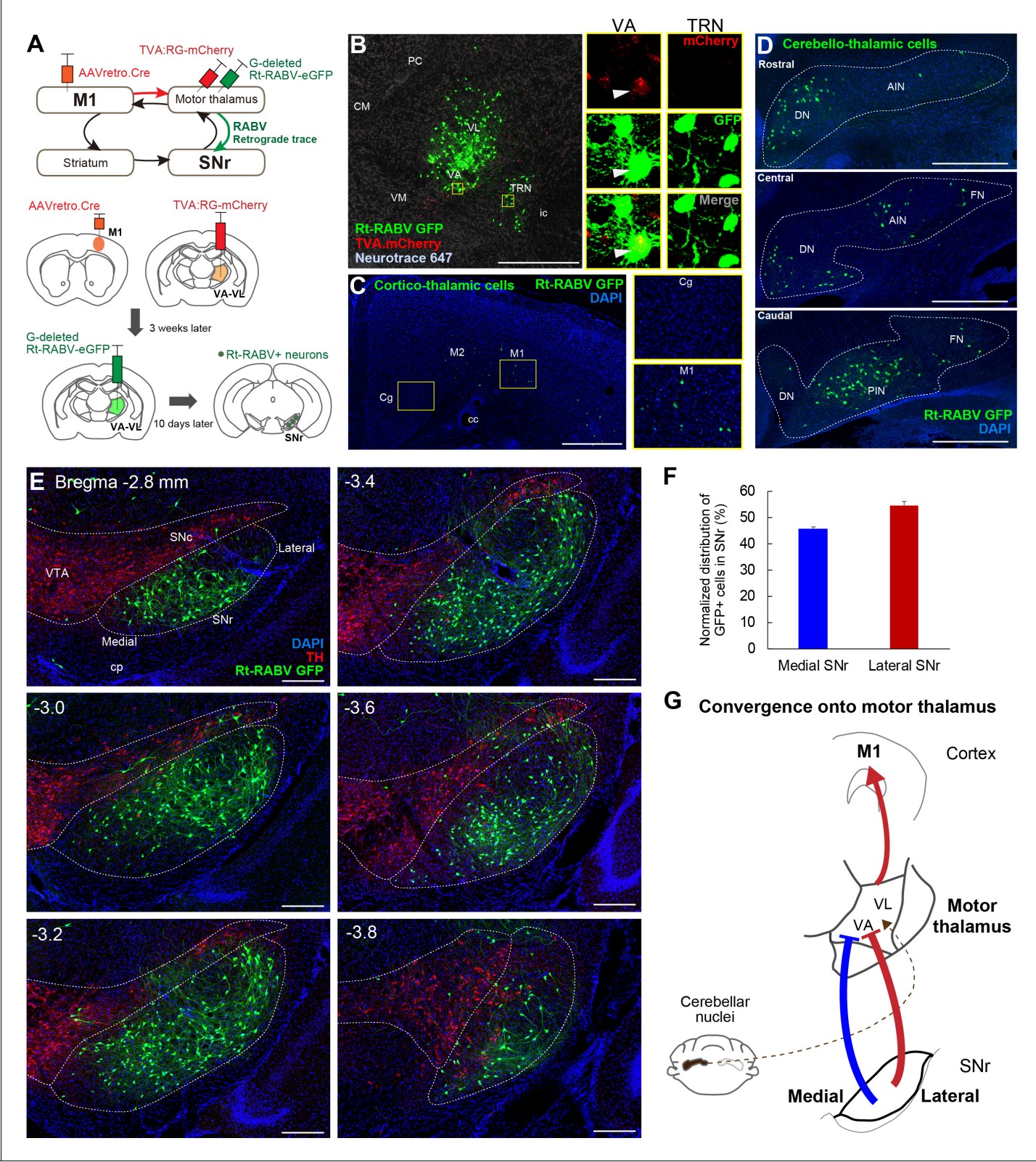

**Figure 3.** Subcircuit-specific modified rabies tracing reveals the convergence of synaptic inputs from medial and lateral SNr onto thalamo-cortical neurons targeting M1. (**A**) Strategy for identifying SNr neurons that synapse onto thalamo-cortical cells projecting to M1. (**B**) Images of the starter cell population (mCherry+/GFP+) in VA-VL motor thalamus and trans-synaptically labeled GFP+ cells in TRN (mCherry-). White arrowheads in images for VA indicate a representative starter cell. Scale bar, 500 μm. (**C**) Images of Rt-RABV-GFP+ cortico-thalamic cells specifically located in ipsilateral motor

*Figure 3 continued*

cortex, but not in cingulate cortex (Cg). Scale bar, 500 µm. (D) Images of Rt-RABV-GFP+ cerebello-thalamic cells mainly located in the contralateral dentate and interpositus nuclei. Scale bar, 500 µm. (E) Images of Rt-RABV-GFP+ nigro-thalamic cells relative to TH+ dopamine cells. Note that both medial and lateral SNr neurons synapse onto motor thalamus. Scale bar, 200 µm. (F) Quantification of the distribution of Rt-RABV-GFP+ cells in medial and lateral SNr (N = 5). Data are expressed as mean ± SEM. (G) Summary diagram showing the convergence of synaptic inputs from medial and lateral SNr to motor thalamus. Cerebellar convergence is also shown as a dashed line. Abbreviations: CM, centromedial; PC, paracentral; CL, centrolateral; VM, ventromedial thalamus; TRN, thalamic reticular nucleus; Cg, cingulate cortex; DN, cerebellar dentate nucleus; AIN, anterior interpositus nucleus; FN, fastigial nucleus; PIN, posterior interpositus nucleus.

DOI: https://doi.org/10.7554/eLife.49995.009

The following figure supplement is available for figure 3:

**Figure supplement 1.** Detailed analysis of the distribution of Rt-RABV-GFP cells in SNr after motor thalamus injection.

DOI: https://doi.org/10.7554/eLife.49995.010

nuclei, including mediodorsal (MD), paracentral and centrolateral (PC/CL), VA-VL, and ventromedial (VM) nuclei (*Figure 4H and I*), consistent with earlier studies (*Cebrián et al., 2005*; *Deniau and Chevalier, 1992*; *Kuramoto et al., 2011*; *Sakai et al., 1998*). More careful analysis revealed that lateral SNr, which receives input from DLS, projects to VA-VL and the dorsal part of caudal VM. By contrast, medial SNr, which receives limbic input from VS, has more diffuse projections and targets MD, VA, rostral VM, and the ventral part of caudal VM. As SNr neurons also express the GABA transporter VGAT (*Rossi et al., 2016*), we subsequently performed Cre-dependent AAV tracing from medial SNr of VGAT-Cre mice and confirmed that medial SNr projections target VA, rostral VM, and the ventral part of caudal VM (*Figure 4—figure supplement 1F–H*).

We next examined thalamic neurons projecting to mPFC, M2 and M1 by simultaneous triple retrograde tracing from each cortical region using fluorescent-conjugated CTb (*Figure 4J–4L* and *Figure 4—figure supplement 1E*). M1- and M2-projecting neurons were found in VA-VL and caudal VM, whereas mPFC-projecting thalamic neurons were selective to rostral VM and MD (*Figure 4J–4L*), consistent with previous reports (*Collins et al., 2018*; *Hunnicutt et al., 2014*; *Kuramoto et al., 2015*). In conjunction with the projection patterns from medial SNr, these data suggest that medial SNr projects preferentially to mPFC-projecting 'limbic' thalamus (rostral VM and MD), but also targets M1- and M2-projecting VA and caudal VM 'motor' thalamus (*Figure 4H and I* and *Figure 4—figure supplement 1F–H*), consistent with identified synaptic inputs from medial SNr to thalamo-M1 cells (*Figure 3*). Therefore, a critical node at which limbic output splits into motor circuits is formed by the medial SNr projections to these regions of motor thalamus (*Figure 4M and N*), through which the limbic loop can exert its unidirectional influence over the motor loop. Conversely, the lateral SNr does not project to mPFC-projecting thalamic nuclei (*Figure 4G–4L*), consistent with the lack of DLS connections to mPFC and suggesting no motor-to-limbic interactions through the direct pathway (*Figure 1*).

To test the functional strength of the projection from VS onto motor thalamus-projecting SNr neurons, we next performed ex vivo patch-clamp recordings of SNr neurons while stimulating striatal terminals within SNr (*Figure 4O–4R*). Here, retrobeads were injected into motor thalamus to label SNr neurons retrogradely and AAV.hsyn.ChR2.eYFP was injected into either DLS or VS for optical stimulation of their terminals in SNr. Retrobead placement in motor thalamus was verified by stereotaxic coordinates and preferential retrograde labeling of cortico-thalamic cells in M1 (*Figure 4—figure supplement 1I and J*). In brain slices, optogenetic activation of DLS terminals in SNr evoked inhibitory post-synaptic currents (IPSCs) in retrobeads+ neurons in lateral SNr, confirming that DLS functionally synapses onto motor thalamus-projecting SNr neurons (*Figure 4O*). Notably, optogenetic activation of VS terminals in SNr also evoked IPCSs in retrobeads+ neurons in medial SNr (*Figure 4P*), confirming that projections from VS to SNr are functional. Comparison of both the IPSC amplitude (*Figure 4Q*) and the paired-pulse ratio (*Figure 4R*) revealed no significant difference in the strength or general properties between VS- and DLS-synapses onto motor thalamus-projecting SNr cells, suggesting that both VS and DLS can equally modulate SNr activity.

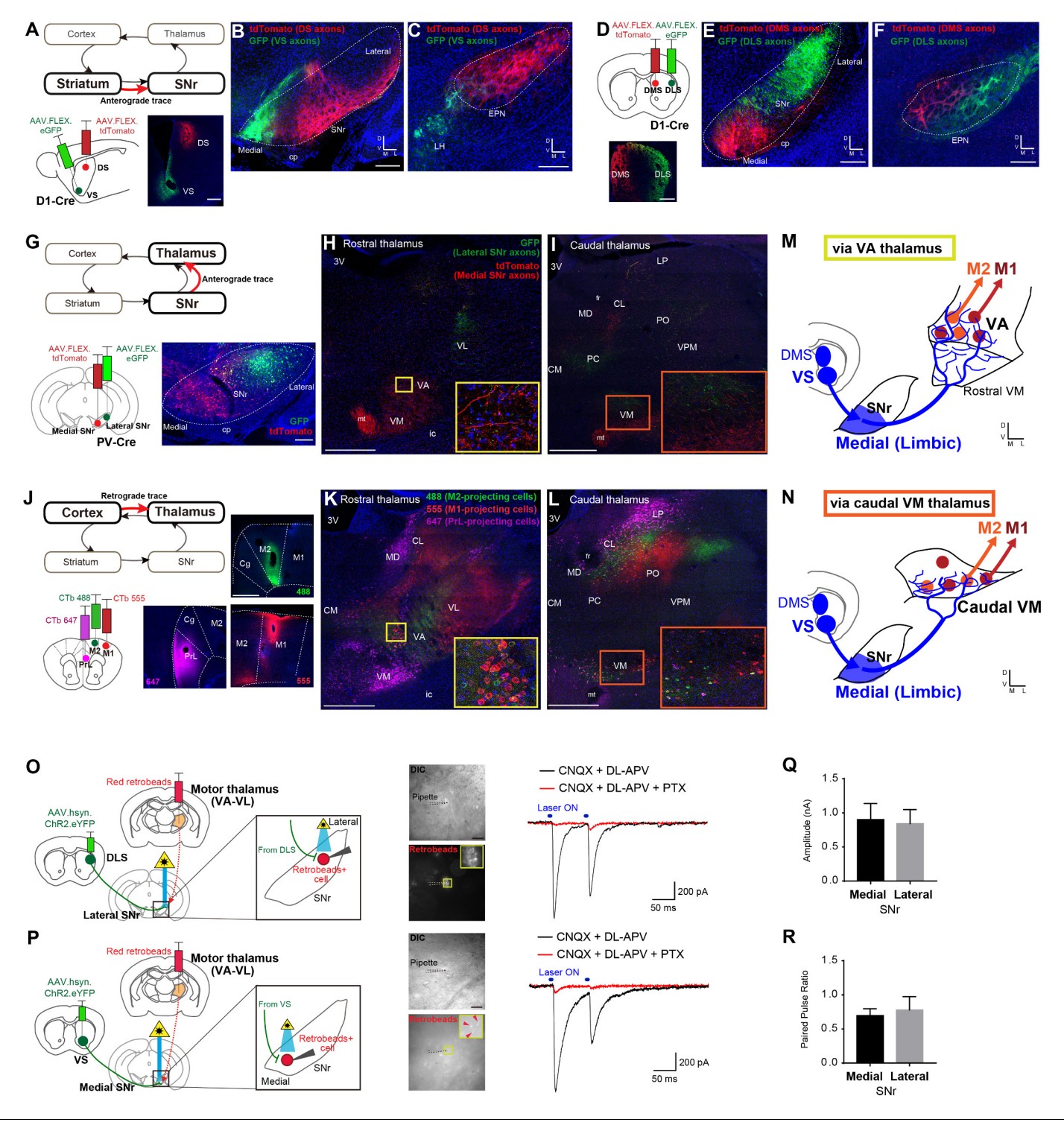

**Figure 4.** Ventral striatum – medial SNr – motor thalamus circuitry provides a mechanism underlying limbic-to-motor connectivity through the direct pathway. (A–C) Cre-dependent, AAV anterograde tracing of striato-nigral projections from VS and DS using D1-Cre mice (A) shows segregated terminal fields in SNr (B), in which VS innervates the medial SNr. DS and VS innervate EPN and LH, respectively (C). N = 3, including CTb-based tracing. Scale bars: 500 μm (A) and 200 μm (B, C). (D–F) The same strategy as in panels (A–C) for mapping DMS versus DLS efferents. N = 3, including CTb-based tracing. Scale bars: 500 μm (D) and 200 μm (E, F). (G–I) Dual Cre-dependent AAV anterograde tracing of nigro-thalamic projections using PV-Cre mice. Conjugated GFP and tdTomato are separately expressed in the lateral and medial SNr (G), which then target distinct thalamic nuclei (H, I). Insets emphasize the medial SNr projections to VA and to the caudal VM. Scale bars: 200 μm (G) and 500 μm (H,I). (J–L) Thalamo-cortical neurons projecting

*Figure 4 continued on next page*

*Figure 4 continued*

to mPFC, M2 and M1 were identified using three fluorophores of CTb. Insets in (**K, L**) indicate VA and caudal VM thalamus that contain M1- and M2-projecting thalamic neurons, which correspond to the insets in (**H**) and (**I**) where medial SNr sends axon terminals. Scale bars, 500 µm. (**M, N**) Schematic illustration showing that medial SNr, which receives VS and DMS inputs, projects to VA and caudal VM thalamus, which in turn project to M1 and M2, as the mechanism for limbic-to-motor connectivity through the direct pathway. (**O, P**) Ex vivo electrophysiology is used to determine the functional strength of inputs from DLS or VS onto SNr neurons projecting to motor thalamus. Injections of red fluorescent protein retrobeads in VA-VL and AAV. hsyn.ChR2.eYFP in the striatum resulted in striato-nigral labeled axons and retrogradely labeled retrobeads+ cells in the SNr (left). Images indicate recording pipettes attached to retrobeads+ cells (middle). Example traces recorded from lateral (**O**) and medial SNr neurons (**P**) under optical stimulation (blue bars above traces, right). Under the glutamate receptor antagonists (CNQX and DL-APV), IPSCs are visible immediately after stimulation, which were abolished by application of picrotoxin (PTX) that blocks GABAa receptors. Scale bars: 20 µm. (**Q, R**) Mean amplitude (medial, 0.902 ± 0.235; lateral, 0.842 ± 0.206; unpaired t-test, $t_{11}$ = 0.177, p=0.862) and paired pulse ratio (medial, 0.697 ± 0.099; lateral, 0.778 ± 0.196; unpaired t-test, $t_{11}$ = 0.410, p=0.690) for all recorded neurons in medial (n = 8 from seven mice) and lateral SNr (n = 5 from two mice). There is a neuron in medial SNr that did not respond to optical stimulation, and we excluded it from the analysis. Note that this neuron was located outside of a ChR2.YFP positive area. Data presented as mean ± SEM. Abbreviations: cp, cerebral peduncle; ic, internal capsule; mt, mammillothalamic tract; fr, fasciculus retroflexus; MD, mediodorsal; CM, centromedial; PC, paracentral; CL, centrolateral; LP, lateral posterior; PO, posterior; VPM, ventroposterior medial thalamus; PrL, prelimbic cortex; Cg, cingulate cortex.

DOI: https://doi.org/10.7554/eLife.49995.011

The following source data and figure supplement are available for figure 4:

**Source data 1.** Source data for ex vivo slice electrophysiology.
DOI: https://doi.org/10.7554/eLife.49995.013

**Figure supplement 1.** Detailed analyses of injection sites for viral tracing for basal ganglia topography.
DOI: https://doi.org/10.7554/eLife.49995.012

## In vivo optogenetic stimulation confirms that ventral striatum controls motor cortex

To determine the ability of DLS and VS to modulate cortical activity, we next recorded mPFC and M1 neurons in vivo with optogenetic stimulation of DLS or VS terminals in SNr (*Figure 5*). Both anterograde and retrograde tracing of striato-nigral projections confirmed that inputs to medial and lateral SNr come from segregated populations in VS/DMS, and DLS, respectively (*Figure 4A–F* and *Figure 5—figure supplement 1A*), allowing us to target their outputs individually using optogenetics. We therefore made AAV.hsyn.ChR2.eYFP injections into the DLS or VS of different mice and then optogenetically activated DLS terminals in the lateral SNr or VS terminals in the medial SNr (*Figure 5B and C*), while recording from mPFC and M1.

We first analyzed the response latency of all the cortical neurons that significantly changed their firing rates during stimulation, and found that the peak in the latency distribution was less than 40 ms (*Figure 5—figure supplement 1B*), consistent with the fastest route only containing three synapses in the striato-nigro-thalamo-cortical pathway. One technical concern is the possibility of AAV. hsyn.ChR2.eYFP virus spreading to the cortex upon injection into the striatum, which may have allowed antidromic activation of the cortical neurons via cortico-nigral projections or cortical axons through SNr (*Naito and Kita, 1994*). We therefore set a lower limit of 5 ms to remove any potential antidromic response of cortical neurons to the optical stimulation (*Li et al., 2015*). Furthermore, we performed additional electrophysiological recording from each downstream target of striatum, including SNr, motor thalamus, and M1, while optogenetically activating dopamine receptor type-1 (D1R) expressing spiny projection neurons in DLS, using D1-Cre mice expressing ChR2 in the striatum, to further characterize the range of response latencies (*Figure 5—figure supplement 2*). On the basis of these results and the previously identified synaptic properties of the striato-nigro-thalamo-cortical circuitry (*Beurrier et al., 2006*; *Cruikshank et al., 2007*; *Kase et al., 2015*; *Lalive et al., 2018*), we have thus restricted our analyses to cortical neurons whose response occurred between 5–35 ms after striatal terminal stimulation (*Figure 5—figure supplement 1B*). Please note that this rather strict criterion was employed in order to avoid any false positives, as the response after 35 ms might be susceptible to network effects, either within the cortical local circuitry or beyond the nigro-thalamo-cortical pathway.

We found that 24% (44/189 recorded cells) of neurons recorded in M1 were responsive to the activation of DLS terminals in SNr (*Figure 5D–G*). Consistent with previous observations (*Lee et al., 2016*; *Oldenburg and Sabatini, 2015*), the activation of the striatal direct pathway mostly increased

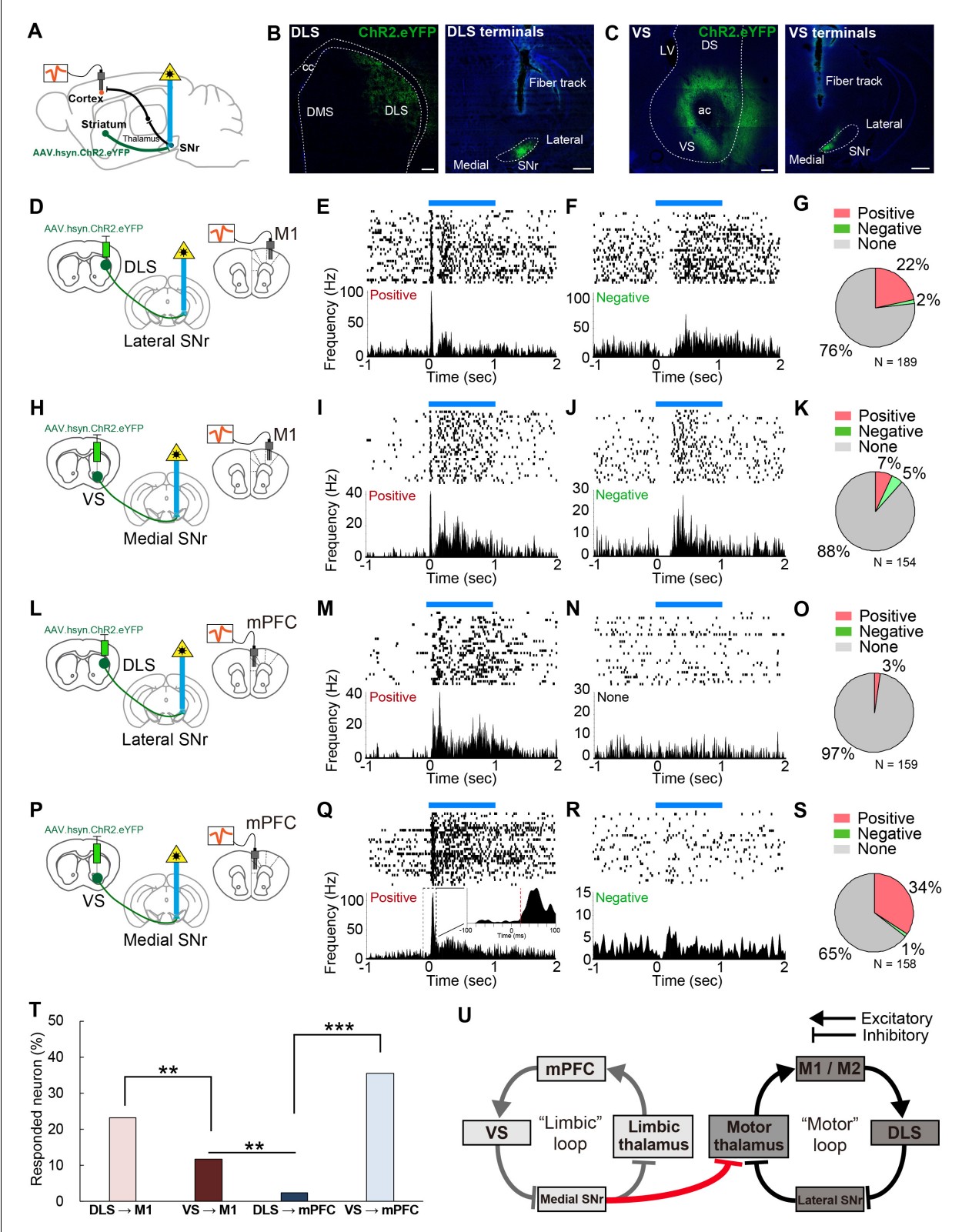

**Figure 5.** In vivo optogenetic stimulation of VS terminals in SNr modulates M1 activity, but DLS terminal stimulation in SNr does not affect mPFC activity. (A) A schematic diagram for in vivo multi-unit recording with optical stimulation. AAV.hsyn.ChR2.eYFP was injected in DLS or VS. During recording, an optic fiber was implanted into either medial or lateral SNr and the recording electrodes were placed in M1 and mPFC. The order of recording from M1 and mPFC was counter-balanced. (B, C) Sample images of the expression of AAV.hsyn.ChR2.eYFP in DLS (B) or VS (C), and their

*Figure 5 continued*

terminals in SNr. Location of terminal labeling was consistent with viral tracing experiments (*Figure 4A–F*). Scale bars: 200 µm (striatum) and 500 µm (SNr). (D–G) In vivo recording from M1 with optical activation of DLS terminals in SNr (n = 189 cells from three mice). Schematic diagram for this experiment (D). (E, F) Examples of positively (E, excited between 5–35 ms) and negatively responding neurons (F, inhibited between 5–35 ms). The upper and lower panels show raster plots, and peri-stimulus time histograms (PSTHs), respectively. Blue horizontal bars indicate 1 s optogenetic stimulation. (G) Pie charts indicating the proportion of positive, negative and unresponsive neurons within the time window (5–35 ms after the onset of optical stimulation), respectively. (H–K) Same as above with the condition of VS terminal stimulation with M1 recording (n = 154 cells from four mice). (L–O) Same as above with the condition of DLS stimulation with mPFC recording (n = 159 cells from three mice). (P–S) Same as above with the condition of VS stimulation with mPFC recording (n = 158 cells from four mice). The inset in (Q) is a zoom-in panel showing an example of the detected latency (red dashed line) based on a change in activity beyond 3SD from baseline activity (see details in Materials and methods). (T) A comparison of the percentage of responded cortical neurons to optogenetic stimulation. These results indicate that VS axonal stimulation alters the activity of M1, but that DLS stimulation rarely evokes responses in mPFC. Z-tests: DLS to M1 vs VS to M1, z = 2.775, p=0.0055; VS to M1 vs DLS to mPFC, z = 3.174, p=0.0015; and DLS to mPFC vs VS to mPFC, z = 7.483, p<0.0001. (U) Schematic diagram of the present study unveiling a unidirectional limbic-to-motor connection between limbic and motor cortico-basal ganglia-thalamocortical loops. This VS - medial SNr - motor thalamus circuitry involves classic direct pathway disinhibition, thus driving increases in activity in the downstream cortical target, as shown in our in vivo electrophysiology.

DOI: https://doi.org/10.7554/eLife.49995.014

The following source data and figure supplements are available for figure 5:

**Source code 1.** Matlab code for in vivo recording data analysis, including the raster and PETH plot.

DOI: https://doi.org/10.7554/eLife.49995.017

**Figure supplement 1.** Additional analyses of in vivo responses of cortical neurons to optogenetic stimulation of VS- or DLS-terminals in SNr.

DOI: https://doi.org/10.7554/eLife.49995.015

**Figure supplement 2.** Electrophysiological recording from neurons in SNr, motor thalamus, and M1 while stimulating D1R direct pathway neurons in DLS and VS.

DOI: https://doi.org/10.7554/eLife.49995.016

cortical neuron firing rates (*Figure 5E and G*), rather than inhibiting the firing of these neurons (*Figure 5F*). Stimulation at different frequencies did not change neuronal response properties (*Figure 5—figure supplement 1F and G*). Strikingly, 12% (18/154) of M1 neurons also responded to VS terminal activation, demonstrating that VS output alters the activity of M1 in vivo (*Figure 5H–K*). Conversely, mPFC neurons responded selectively to the stimulation of VS terminals in medial SNr (35%, 56/158), but not to DLS terminal stimulation (3%, 4/159) (*Figure 5L–S*). These differential effects did not result from variability of optic fiber placement and virus expression, as the mPFC and M1 responses were both recorded in the same animal, using a single optic fiber fixed in either medial or lateral SNr and under the same stimulation parameters. Comparison of the proportion of neurons that changed their activity in response to optical stimulation indicated a significantly greater portion of M1 neurons responding to VS stimulation than of mPFC neurons that were responsive to DLS stimulation (*Figure 5T*, p=0.0015). Notably, using more strict (5–15 or 5–25 ms) or loose (5–45 ms) criteria did not change the overall proportions of responsive neurons (*Figure 5—figure supplement 1C–E*). Moreover, we tested the effect of activation of the VS direct pathway neurons (D1R neurons) on M1 activity using D1-Cre mice (*Figure 5—figure supplement 2P–T*). This indicated that 8% of M1 neurons responded to the VS activation with a latency of 5–35 ms, further supporting the results of our terminal stimulation experiments. These findings provide in vivo evidence for the existence of a one-way limbic-to-motor interaction in which the ventral striatum exerts a unidirectional influence over motor cortex, in addition to the strong modulation within each functional loop (*Figure 5U*).

## Discussion

In this study, we dissected the topography of limbic and motor cortico-basal ganglia loops at each synaptic step and revealed a one-way influence of limbic loops onto motor loops. Cortico-basal ganglia circuitry has been considered to consist mostly of parallel, segregated loops within different functional domains (*Alexander et al., 1986*; *Haber, 2003*; *Kelly and Strick, 2004*; *Kim and Hikosaka, 2015*; *Miyachi et al., 2006*; *Parent and Hazrati, 1995*), with a possibility of some open-loop architecture providing interactions between domains (*Haber, 2003*; *Joel and Weiner, 1994*; *Kelly and Strick, 2004*; *Miyachi et al., 2006*). Although our data do demonstrate mostly closed cortico-basal ganglia loops within each domain, our results confirm an open cortico-basal ganglia loop

that allows for a one-way interaction from limbic to motor circuitry (*Figure 5U*). The open-loop structure that we revealed here is consistent with earlier studies in primates that have identified multi-synaptic connectivity from ventral putamen to M1 (*Kelly and Strick, 2004*; *Miyachi et al., 2006*), with conceptual work on the convergence of basal ganglia outputs to motor circuits (*Allen and Tsukahara, 1974*; *Haber, 2003*; *Joel and Weiner, 1994*; *Kemp and Powell, 1971*), and with behavioral findings that suggest the involvement of VS in modifying motor output (*Belin and Everitt, 2008*; *Floresco, 2015*; *Sawada et al., 2015*). In the present study, we confirmed this limbic-to-motor connectivity by trans-synaptic rabies tracing, two distinct circuit-specific monosynaptic rabies tracing experiments (nigro-thalamic and thalamo-cortical), and viral tracing combined with transgenic mice. Subsequently, we demonstrated its functionality by ex vivo slice recordings, which physiologically showed inhibition of motor-thalamus projecting SNr neurons by VS inputs. We further carried out in vivo electrophysiology recordings that verified functional closed loops within VS-mPFC and DLS-M1, and that also found significant activation of M1 by VS inputs to SNr, in contrast to the absence of modulation of mPFC by the DLS inputs to SNr. These findings suggest considerable functional influence of limbic striatal output on M1 activity and reveal a more complex framework of interactions across functionally distinct cortico-basal ganglia loops than is widely appreciated. In addition to VS, the other striatal territories, including DMS and TS, were also found to connect to the motor cortex, suggesting a wider theme of all modalities converging onto motor circuits, in addition to their within-domain interactions. Unlike the topographically segregated cortico-striatal projections, we propose that striatal outputs throughout basal ganglia are motor-oriented in nature while maintaining domain-dependent closed loops, likely structured to be able to drive behavioral output in various contexts determined by the inputs.

Our results provide important insights into how the non-motor regions of the striatum, including VS, DMS and TS, might influence motor cortex to modulate behavioral output. These striatal subregions are each associated with different aspects of behavioral control (*Floresco, 2015*; *Ito and Doya, 2015*; *Yin et al., 2004*; *Yin et al., 2005*). For instance, VS has been suggested to be involved in Pavlovian approach (*Saunders and Robinson, 2012*) and in emotion-driven actions such as avoidance behavior (*Ramirez et al., 2015*). Notably, a recent study in primates has found that VS plays a crucial role in the recovery of skilled movement after spinal cord injury, by driving neural activity in primary motor cortex (*Sawada et al., 2015*). Our results offer a mechanism through which limbic information originating from ventral striatum can influence motor cortex and motor output via the nigro-thalamic pathway, providing an anatomical foundation that is capable of supporting these behavioral changes. Overall, a unidirectional limbic-to-motor influence through this pathway implies that selecting and invigorating action can be determined by emotional and motivational states. Our findings also have important implications for the DMS and TS, which are associative and sensory regions of the striatum, respectively. The DMS receives cognitive information from higher-order association cortex and is known to be crucial for internally driven, goal-directed behavior (*Kupferschmidt et al., 2017*; *Yin et al., 2009*). On the other hand, TS has been thought to primarily process various modalities of sensory information from the external world (*Alloway et al., 2017*; *Hintiryan et al., 2016*). Emerging evidence suggests that TS is involved in responses to novel or salient stimuli (*Menegas et al., 2018*) and in sensory-guided action selection (*Znamenskiy and Zador, 2013*). In the present study, we uncovered a mechanism through which the DMS and TS can interact with motor cortex, enabling information involving internal and external states to affect motor circuits for guiding motor output.

A remaining question is how and where limbic and motor information are integrated throughout basal ganglia regions, on their way towards motor cortex. We obtained anatomical and physiological evidence that medial SNr, which receives input from VS, can target motor thalamus and thus motor cortex, through which the limbic information is conveyed to motor circuits. Yet, it does not rule out integration at other levels. It is plausible that the limbic and motor integration occurs as early as SNr, as it has been known that single SNr neurons have dendrites that extend broadly across the medio-lateral extent of SNr (*Mailly et al., 2001*). In fact, prior studies showed that SNr neurons receive functionally distinct inputs from striatum and other indirect pathway nuclei (*Bevan et al., 1994*; *Bevan et al., 1996*). In this respect, our results showing that the absence of tri-synaptic connections from DLS to mPFC as well as from VP to M1 is notable, as they are separated from such synaptic- or circuit-level integration despite their projections to SNr (*Deniau et al., 1996*; *Tripathi et al., 2013*). Regarding the topography of axon terminals of the striatal-nigral pathway,

our choice of two extremes (limbic and motor) revealed no overlap of the terminal fields in SNr (*Figure 4A–4F*). Note that such an overlap of striatal axons has been shown to occur if originating from neighboring regions of striatum (*Deniau et al., 1996*) as opposed to distant sites, such as those targeted in our current study (*Figure 4*). In addition, interactions across cortico-basal ganglia-thalamocortical loops are also enabled by dopamine modulation with the divergent projections from the ventral tegmental area (VTA) and substantia nigra pars compacta (SNc) targeting both dorsal and ventral striatum (*Beier et al., 2015*; *Haber et al., 2000*; *Lerner et al., 2015*; *Watabe-Uchida et al., 2012*; *Yang et al., 2018*). Elucidating the functional significance of these other potential mechanisms underlying cross-modal integration onto motor loops will yield deeper insights into the behavioral relevance of the partially open looped architecture of the cortico-basal ganglia system.

In summary, we provide evidence for an interaction across functionally distinct cortico-basal ganglia loops, allowing limbic information to affect motor circuits by ventral striatum control of motor cortex through basal ganglia direct pathway output. These findings pave the way for a more complete understanding of fundamental aspects of behaviors such as action sequencing and habit formation (*Dickinson, 1985*; *Jin and Costa, 2015*; *Yin and Knowlton, 2006*), and have important implications in a wide range of neurological and psychiatric diseases, from obsessive-compulsive disorder (OCD) to anxiety and depression, in which the limbic control of action is compromised (*Everitt and Robbins, 2005*; *Marchand et al., 2012*; *Redgrave et al., 2010*; *Robbins et al., 2012*; *Vaghi et al., 2017*).

### Contact for reagent and resource sharing

Requests for research materials should be directed to the Lead Contact, Xin Jin (xjin@salk.edu).

## Materials and methods

**Key resources table**

| Reagent type (species) or resource | Designation | Source or reference | Identifiers | Additional information |
|---|---|---|---|---|
| Strain, strain background (*Rattus norvegicus*) | Wistar rats | Chales River, France | Wistar IGS rats, strain code: 003 | |
| Strain, strain background (*Mus musculus*) | Wild-type C57BL/6 mice | Jackson Laboratory | JAX:000664 | |
| Strain, strain background (*Mus musculus*) | *Drd1a-Cre* mice | MMRRC | RRID: MMRRC_034258-UCD | |
| Strain, strain background (*Mus musculus*) | *Pvalb-Cre* mice | Jackson Laboratory | JAX:008069 | |
| Strain, strain background (*Mus musculus*) | *Slc32a1-Cre (VGAT-Cre)* mice | Jackson Laboratory | JAX:016962 | |
| Strain, strain background (*Mus musculus*) | Ai32 mice (B6;129S-Gt(ROSA) 26Sor$^{tm32(CAG-COP4* H134R/EYFP)Hze}$/J) | Jackson Laboratory | JAX:012569 | |
| Strain, strain background (Adeno-associated virus) | AAV5/EF1-Flex-TVA-Cherry | UNC Viral Vector Core | RRID: SCR-002448 | |
| Strain, strain background (adeno-associated virus) | AAV8/CA-Flex-RG | UNC Viral Vector Core | RRID: SCR_002448 | |

*Continued on next page*

*Continued*

| Reagent type (species) or resource | Designation | Source or reference | Identifiers | Additional information |
|---|---|---|---|---|
| Strain, strain background (recombinant rabies virus) | EnvA.dGRabies.eGFP | Salk Vector Core | RRID: SCR_014847 | |
| Strain, strain background (adeno-associated virus) | AAV9.FLEX.tdTomato | University of Penn Viral Vector Core | RRID: SCR_015406 | |
| Strain, strain background (adeno-associated virus) | AAV9.FLEX.eGFP | University of Penn Viral Vector Core | RRID: SCR_015406 | |
| Strain, strain background (adeno-associated virus) | AAV9.CAG.tdTomato | UNC Viral Vector Core | RRID: SCR_002448 | |
| Strain, strain background (adeno-associated virus) | AAV9.hsyn.ChR2.eGFP | University of Penn Viral Vector Core | RRID: SCR_015406 | |
| Strain, strain background (adeno-associated virus) | AAV5-EF1a-DIO-hChR2(H134R)-mCherry | University of Penn Viral Vector Core | | |
| Antibody | anti-wild type rabies phosphoprotein mouse monoclonal antibody | commercially unavailable (*Raux et al., 1997*) | | (1:5000) |
| Antibody | anti-cholera toxin b-subunit goat polyclonal antibody | List Biological Laboratories | Cat.# 704 | (1:15000) |
| Antibody | anti-tyrosine hydroxylase (TH) mouse monoclonal antibody | Millipore | Cat.# MAB318 | (1:1000) |
| Antibody | anti-NeuN rabbit polyclonal antibody | Abcam | Cat.# ab104225 | (1:1000) |
| Antibody | anti-GFP chicken polyclonal antibody | Novus Biologicals | Cat.# NB100-1614 | (1:1000) |
| Antibody | anti-substance P mouse monoclonal antibody | Abcam | Cat.# ab14184 | (1:1000) |
| Antibody | anti-mouse IgG horseradish peroxydase (HRP) (host: rabbit, polyclonal) | DAKO | Cat.# P260 | (1:200) |
| Antibody | anti-goat IgG horseradish peroxydase (HRP) (host: rabbit, polyclonal) | DAKO | Cat.# P044901-2 | (1:200) |
| Antibody | anti-mouse Alexa Fluor 488 (host: donkey, polyclonal) | Jackson ImmunoResearch Laboratories | Cat.# 715-545-151 | (1:250) |
| Antibody | anti-mouse Cy3 (host: donkey, polyclonal) | Jackson ImmunoReseach Laboratories | Cat.# 715-165-151 | (1:250) |
| Antibody | anti-mouse Cy5 (host: donkey, polyclonal) | Jackson ImmunoReseach Laboratories | Cat.# 715-175-151 | (1:250) |

*Continued*

| Reagent type (species) or resource | Designation | Source or reference | Identifiers | Additional information |
|---|---|---|---|---|
| Antibody | anti-rabbit Alexa Fluor 488 (host: donkey, polyclonal) | Jackson ImmunoReseach Laboratories | Cat.# 711-545-152 | (1:250) |
| Antibody | anti-rabbit Cy3 (host: donkey, polyclonal) | Jackson ImmunoReseach Laboratories | Cat.# 711-165-152 | (1:250) |
| Antibody | anti-rabbit Cy5 (host: donkey, polyclonal) | Jackson ImmunoReseach Laboratories | Cat.# 711-175-152 | (1:250) |
| Antibody | anti-chicken Alexa Fluor 488 (host: donkey, polyclonal) | Jackson ImmunoReseach Laboratories | Cat.# 703-545-155 | (1:250) |
| Chemical compound, drug | Neurotrace 647 | Invitrogen | Cat.# N21483 | (1:250) |
| Chemical compound, drug | DL-APV | Tocris | Cat.#. 0106 | |
| Chemical compound, drug | CNQX | Sigma-Aldrich | Cat.#. C239 | |
| Chemical compound, drug | Picrotoxin | Sigma-Aldrich | Cat.#. P1675 | |
| Chemical compound, drug | QX-314 | Sigma-Aldrich | Cat.#. L5783 | |
| Chemical compound, drug | cholera-toxin b-subunit in low-salt | List Biological Laboratories | Cat.# 104 | |
| Chemical compound, drug | cholera-toxin b-subunit Alexa Fluor 488 conjugate | Invitrogen | Cat.# C22841 | |
| Chemical compound, drug | cholera-toxin b-subunit Alexa Fluor 555 conjugate | Invitrogen | Cat.# C34776 | |
| Chemical compound, drug | cholera-toxin b-subunit Alexa Fluor 594 conjugate | Invitrogen | Cat.# C22842 | |
| Chemical compound, drug | cholera-toxin b-subunit Alexa Fluor 647 conjugate | Invitrogen | Cat.# C34778 | |
| Chemical compound, drug | Red Retrobeads | Lumafluor Inc | | |
| Software, algorithm | Matlab | MathWorks | R2015b | |
| Software, algorithm | Prism | GraphPad | GraphPad Prism 7 | |
| Software, algorithm | Neurolucida | MBF Bioscience | NL-11 | |
| Software, algorithm | CorelDRAW | Corel | CorelDRAW Graphics Suite X7 | |
| Software, algorithm | ImageJ | NIH | ImageJ Win64 bit | |
| Software, algorithm | ZEN | Zeiss | ZEN | |
| Software, algorithm | pClamp 9.2 | Molecular Devices | Molecular Devices | |
| Software, algorithm | Offline Sorter | Plexon | Version 3.3.3 | |

All procedures related to trans-synaptic wild-type rabies tracing were carried out in accordance with the European guidelines for the care and use of laboratory animals and with the guidelines of the French Ministry for Agriculture and Fisheries, Division of animal rights. They were approved by the ethics committee in Neuroscience at the INT (nr. 02167.01). Recombinant monosynaptic rabies, viral and other tracing experiments were conducted at the Salk Institute for Biological Studies according to NIH guidelines, and all protocols were approved by their Institutional Animal Care and Use Committee. All the experimenters handling rabies virus were vaccinated before handling.

## Animals

Male Wistar rats were used for trans-synaptic wild-type rabies experiments (*Aoki et al., 2019*). All other experiments were performed in mice maintained on a C57BL/6 background, except for *Slc32a1(VGAT)-Cre* (mixed with C57BL/6 and 129/Sv). Male and female wild-type mice were used for monosynaptic rabies tracing, non-Cre-dependent virus tracing, and for other tracing experiments that did not require specific Cre-lines. In the cell-type specific tracing, *Drd1a-Cre*, *Pvalb-Cre*, and *Slc32a1(VGAT)-Cre* mice were used. For electrophysiological recording, we used wild-type and *Drd1a-Cre* mice.

## Wt-RABV tracing

For the present trans-synaptic tracing (*Figure 1*), nine out of eleven cases injected were used from the previous study, in which we focused on cerebellar connections into sensorimotor cortices (*Aoki et al., 2019*). Before surgery, animals were anesthetized with ketamine (80 mg/kg, Imalgene, France) and xylazine (10 mg/kg, Bayer, Germany). Appropriate levels of anesthesia were monitored by the absence of whisker movements and foot-pinch withdrawal reflex. Additional doses of the ketamine-xylazine mixture were administered i.p. when necessary. After placing animals into a stereotaxic frame (David Kopf Instruments), a mixture (0.15 or 0.2 µL) of the 'French' subtype of CVS-11 rabies virus (Wt-RABV, *Aoki et al., 2019*; *Coulon et al., 2011*; *Raux et al., 1997*; *Ruigrok et al., 2008*; *Suzuki et al., 2012*; *Ugolini, 2010*) and cholera-toxin b subunit (CTb, low salt; List Biological Laboratories, 1% w/v in 0.2 M phosphate buffer (PB) at pH 7.4: the injection solution consisted of four parts Wt-RABV and one part CTb) was injected by a 1 µl Hamilton syringe in the following functional areas of the cerebral cortex: primary motor cortex (M1, n = 4), secondary motor cortex (M2, n = 4), and medial prefrontal cortex (mPFC, n = 3, one centered on prelimbic and two centered on cingulate cortex). As shown in our previous study, coordinates of the injection sites were determined by reference to the standard rat brain atlas (*Aoki et al., 2019*; *Paxinos and Watson, 2004*). The viral stock was kept at −80°C until use. This CVS-11 strain of RABV has been confirmed to be transported trans-synaptically in a retrograde direction as well as in the time-dependent manner (*Aoki et al., 2019*; *Kelly and Strick, 2004*; *Ruigrok et al., 2008*; *Suzuki et al., 2012*; *Ugolini, 2010*). No neighboring neurons are infected unless they have synaptic contacts to the already infected cells. Adding CTb to the injection solution enabled accurate determination of the injection site (*Aoki et al., 2019*; *Suzuki et al., 2012*). Upon the injection, the needle was left in place for another 5 min to allow the virus to spread. After surgery, animals were monitored for signs of stress or discomfort. Throughout the course of the experiment, all of the animals were kept in a biohazard safety level two room (BSL-2). Survival time was set at 66–70 hr after viral injections, which has been established to be sufficient for 3$^{rd}$-order labeling without evidence of 4$^{th}$-order labeling when tracing from the rat cerebral cortex (*Aoki et al., 2019*). For a control experiment with shorter survival time, we performed the identical procedure and perfused animals at 58 hr after the Wt-RABV injections. All of the animals were euthanized with a lethal dose of sodium pentobarbital (80 mg/kg, i.p., Nembutal, Libourne, France), and perfused with 0.9% saline followed by 4% paraformaldehyde (PFA) in PB. Brains were extracted and post-fixed in 4% PFA for at least a week to kill the rabies virus completely.

## AAVretro.Cre + EnvA-dG-RABV tracing (Rt-RABV tracing)

For monosynaptic Rt-RABV tracing of nigro-thalamic cells (*Figure 2*), we used male or female wild-type mice (C57BL/6 strain, n = 6). Experiments were performed as previously described (*Smith et al., 2016*). Briefly, after placing the animal into a stereotaxic frame (David Kopf Instruments) under isoflurane anesthesia, a subcutaneous injection of bupivacaine was injected into the

scalp on the midline as local anesthesia before the incision. Anesthetic state was maintained by iso-flurane anesthesia administered via a nosecone (1–1.5% in 1 L/min O$_2$). The goal of this experiment was to identify the striatal projection neurons that synapse onto substantia nigra pars reticulata (SNr) neurons that specifically project to VA-VL motor thalamus. To achieve this, we first injected a 1:1 mixture of AAVretro.Cre and AAV.FLEX.tdTomato into VA-VL thalamus, where the AAV.FLEX.tdTomato served to determine the injection site and thalamo-cortical terminals in the cortex. The labeling of the thalamic injection site and its axonal terminals in the cortex helped us to verify whether the injection was made in VA-VL thalamus and whether it innervated motor cortex. Three of six mice received a mixture of the AAVretro.Cre and AAV.FLEX.tdTomato in VA-VL, and another three received only AAVretro.Cre in the same coordinates in the VA-VL. The injected AAVretro.Cre served to induce Cre-recombinase expression in the VA-VL projecting SNr neurons (*Tervo et al., 2016*). A separate injection of a 1:1 mixture of AAV5/EF1-Flex-TVA-mCherry and AAV8/CA-Flex-RG was made in the SNr during this initial surgery. After three weeks of transfection, the G-deleted RABV virus was subsequently injected into SNr in an angled approach 30° from vertical via the contralateral hemisphere. We injected 0.2 µl of the AAVretro.Cre + AAV.FLEX.tdTomato mixture at 1:1 ratio (or AAVretro.Cre solely) in the VA-VL thalamus (AP: −1.1 mm, ML: 1.0 mm, DV: 3.4 mm, all from bregma or dura), and 0.8 µl of TVA.RG in SNr (AP: −3.3, ML: 1.4, DV: 4.3). The following G-deleted RABV.eGFP was injected in the same site in SNr, where we applied the 30° angled injection from the other hemisphere to avoid undesired contamination of starter cells for monosynaptic rabies tracing. A survival time of 10 days allowed for the successful infection of Rt-RABV. For the control experiment without using rabies glycoprotein, we applied a similar procedure but injected only AAV5/EF1-Flex-TVA-mCherry into SNr with no conjugation of AAV8/CA-Flex-RG. All animals were perfused under ketamine-xylazine anesthesia, and extracted brains were kept in 4% PFA for overnight.

For monosynaptic Rt-RABV tracing of thalamocortical cells targeting M1 (*Figure 3*), we applied the same strategy as above. Briefly, we injected AAVretro.Cre in M1 (0.2 µl in each: AP, +1.6; ML, 1.6; DV, 0.8; AP, +0.6; ML, 1.3; DV, 0.8), and a 1:1 mixture of AAV5/EF1-Flex-TVA-mCherry and AAV8/CA-Flex-RG into VA-VL motor thalamus (0.8 µl: AP, −1.2; ML, 1.0; DV, 3.5). Three weeks later, G-deleted RABV.eGFP (0.8 µl) was injected at the same coordinate of VA-VL motor thalamus. Ten days after the RABV injections, mice were perfused, and their brains were kept in 4% PFA overnight for further histological analysis.

## AAV and CTb tracing

To determine the topography of striato-nigral projections, we used Cre-dependent AAV tracing of striatal D1-type neurons using *Drd1a-Cre mice*. AAV9.FLEX.eGFP (0.4–0.8 µl) and AAV9.FLEX.tdTomato (0.4–0.8 µl) were injected in VS (AP, +1.3; ML, 0.8; DV, 3.9) and DS (AP, +0.8; ML, 2.0; DV, 2.2), respectively. For this VS injection, we applied an angled approach 20° from the rostral cortex to minimize spread of virus to dorsal striatum. The same dual tracing was also performed between DMS (AP, +0.5; ML, 1.5; DV, 2.5), and DLS (AP, +0.5; ML, 2.5; DV, 2.5). Using the same viral tracing strategy, nigro-thalamic projections were determined. Local injections of Cre-dependent AAV were performed in *Pvalb-Cre* and *Slc32a1(VGAT)-Cre* mice. The use of these two transgenic lines allowed us to limit viral transfections to GABAergic neurons in the SNr, so as to make the tracing as clean as possible. We injected 0.1 µl of AAV.FLEX.eGFP or AAV.FLEX.tdTomato in medial (AP, −3.3; ML, 1.0; DV, 4.5) or lateral SNr (AP, −3.3; ML, 1.7; DV, 4.0). The same volume and coordinates were used for retrograde tracing from medial and lateral SNr to the striatum using CTb-647. Three-fluorophore-CTb tracing was conducted with injections in the cortex. In this study, a combination of CTb-488, CTb-555 (or CTb-594), and CTb-647 (Invitrogen) was chosen. Selected injection sites and volumes were as follows: prelimbic cortex (PrL) (0.2 µl: AP, +2.0; ML, 0.3; DV, 1.5), M2 (0.2 µl: AP, +1.4; ML, 0.8; DV, 0.8), M1 (0.2 µl: AP, +1.0; ML, 1.6; DV, 0.8). For the quantification of cortico-striatal projections, we injected non Cre-dependent AAV (YFP or tdTomato) in M1 and PrL in wild-type mice using the injection coordinates mentioned above.

As adequate survival time for each tracing technique, we waited at least 7 days for fluorescent CTb, and 10 days for AAVs until perfusion.

## Histology and immunohistochemistry

For the Wt-RABV tracing, extracted tissue was stored overnight in 10% sucrose in 0.05 M PB in the refrigerator (4°C). The intact brain was embedded in gelatin solution (12% gelatin/10% sucrose in $H_2O$) and sectioned coronally at 40 µm using a freezing microtome (Leica SM 2000R). Serial sections were collected and divided into eight numbered vials. Selected vials were processed with an interval of 160 µm (2 vials out of 8), for either rabies or CTb immunohistochemistry. In rabies immunohistochemistry using 1st and 5th vials, sections were first rinsed with phosphate-buffered saline containing 0.9% NaCl (PBS), and floated in 3% hydrogen oxidase ($H_2O_2$) in PBS for 20 min for blocking reaction against endogenous peroxidase. The sections were incubated overnight at room temperature in an anti-rabies phosphoprotein mouse monoclonal antibody (*Raux et al., 1997*) diluted at 1:5000 in PBS+, that is PBS containing 2% normal horse serum and 0.5% Triton X-100. Subsequently, the sections were incubated in secondary rabbit anti-mouse horseradish peroxidase (Dako, 1:200 in PBS+), followed by the visualization with the incubation in a 3,3'-diaminobenzidine-tetrahydrochloride (DAB) solution (0.025% DAB and 0.005% $H_2O_2$ in 0.05M PB), generating a brown insoluble reaction in rabies-infected neurons. For CTb immunohistochemistry using 2nd and 6th vials, after receiving a similar pre-treatment to reduce endogenous peroxidase reaction, sections were incubated overnight in a polyclonal anti-choleragenoid antibody raised in goat (goat anti-CTb, lot no. 704, List Biological Laboratories) diluted 1:15,000 in PBS+, followed by incubation in biotinylated donkey anti-goat IgG for 90 min (Dako, 1:200, in PBS+). Finally, these sections were visualized by reaction with DAB for 20 min. Upon completion of all steps, the sections were mounted sequentially, Nissl-counterstained by thionin, and cover-slipped with Permount.

For Rt-RABV, AAV and CTb tracing methods, brains were stored in 30% sucrose in 0.1M PB at 4°C before sectioning. We gelatinized brains and sectioned coronally at 50 µm (Microm HM 430, Thermo Scientific), or in a few cases, sectioned brains without gelatin molding. Sections were collected into four vials, with an interval of 200 µm. For Rt-RABV tracing, we used endogenous fluorescence of eGFP for signal detection. For the analysis of starter cells in SNr, we used anti-tyrosine-hydroxylase staining (TH) to delineate dopamine neurons in VTA/SNc. Sections were rinsed in the tris-buffered saline (TBS) three times for 10 min each, followed by 45 min incubation in TBS+ containing 5% normal horse serum and 0.5% Triton X-100. Sections were then incubated in a primary antibody against TH raised in mouse in TBS+ (Millipore, 1:1000) for 48 hr, and then rinsed, incubated in an anti-mouse secondary antibody in TBS+ (Jackson ImmunoReseach Laboratories, 1:250) conjugated with Cy5 for 2 hr. We used the same procedures for the other immunohistochemical staining using primary antibodies including anti-GFP raised in chicken (Novus Biologicals, 1:1000), anti-substance P raised in mouse (Abcam, 1:1000), anti-NeuN raised in rabbit (Abcam, 1:1000) in combinations with appropriate secondary antibody visualization conjugated with either Alexa-488, Cy3 or Cy5 (Jackson ImmunoReseach Laboratories). In each tracing experiment, we selected either DAPI, NeuN, or Neurotrace 647 (Invitrogen) staining for counterstaining, depending on the color availability for microscopy.

## Microscopy and data analysis

Representative brightfield microphotographs were obtained with a digital camera attached to a Keyence microscope (BZ-9000). Representative fluorescent images were taken using a Zeiss LSM 710 laser scanning confocal microscope. When plotting labeled neurons, axon terminals and depicting contours for Wt-RABV tracing, we used an Olympus microscope (BX51W1) equipped with Neurolucida software (MBF Bioscience). For counting and plotting neurons in the other analyses, we examined labeled structures by epi-fluorescent microscope (Axioskop 2, Zeiss), or by Neurolucida offline software (MBF Bioscience) into which the obtained confocal images were imported.

For the Wt-RABV tracing, all the procedures for the identification of the injection sites were described in our previous study (*Aoki et al., 2019*). Briefly, we examined the CTb labeling of the injection site referred to the brain atlas (*Paxinos and Watson, 2004*). In addition, we projected each injection site to a flattened map of the cerebral cortex or three-dimensionally (3D) reconstructed brain. To identify the input-output relationship between striatum and the cerebral cortex, we analyzed the anterogradely labeled CTb+ cortico-striatal terminals and the distribution of Wt-RABV+ neurons in the striatum (*Figure 1*). In this analysis, we selected four different anterior-posterior levels of striatum and plotted CTb+ terminals and Wt-RABV+ striatal neurons using the Neurolucida

software. To detail the pattern of striatal output to the cerebral cortex, we also analyzed the distribution of the Wt-RABV+ neurons across the entire striatum. In this analysis, all Wt-RABV+ neurons in the striatum were counted and plotted. The series of plotted contours and neurons were combined to make a 3D reconstruction of neurons (*Figure 1C*). For the density map of Wt-RABV+ striatal neurons, sections examined for counting and plotting at a 160 μm interval were divided into eight rostro-caudal parts over which multiple sections were overlaid. We then calculated the density of Wt-RABV+ striatal neurons in 160 μm$^2$ bins and made a color-coded density map. To make this map, the most densely labeled bin was first determined, and subsequently all the other bins were normalized relative to the densest bin using a color map in a logarithmic scale. We performed the density-map analysis for every case (only representative cases are shown in *Figure 1*). For quantification of Wt-RABV+ neurons, we put regions of interest (ROIs, a circle with 600 μm of diameter) in five distinct striatal subregions: VMS, VLS, DMS, DLS and TS (*Figure 1*). Using the standard brain atlas (*Paxinos and Watson, 2004*), we determined the approximate center of ROIs for each striatal region as follows (from bregma and dura): VMS (AP, +1.8; ML, 1.1; DV, 6.5); VLS (AP, +1.8; ML, 2.5; DV, 6.5); DLS (AP, +0.7; ML, 3.7; DV, 3.7); DMS (AP, +0.2; ML, 2.3; DV, 4.2); TS (AP, −2.0; ML, 4.7; DV, 5.0). Here, the number of Wt-RABV+ neurons within each ROI was counted in three adjacent sections per animal centered on a section approximately located at the target coordinated mentioned above. Normalized distributions were determined by calculating the percentage of Wt-RABV+ neurons in each striatal sub-region over the total number of Wt-RABV+ neurons found in the five regions. Finally, the normalized distribution of each individual animal was averaged (*Figure 1M*; mPFC, n = 3; M2, n = 4; M1, n = 4). For the analysis of open-loop components (*Figure 1—figure supplement 2G*), we compared the contribution of VS neurons connecting to M1 and M2 (n = 8) and that of DLS neurons connecting to mPFC (n = 3) from their percentages calculated in *Figure 1M*, after averaging VMS and VLS in each case. For the luminance analysis of the cortico-striatal projections, we measured the fluorescent intensity of each M1 and PrL projection to the striatum across five striatal subregions (VMS, VLS, DMS, DLS, and TS) with reference to a previous study (*Nonomura et al., 2018*). Measured luminance was then subtracted by background activity of the fluorescence and normalized across the five subregions. We then averaged percentages in the normalized distributions (N = 2 in each of M1 and PrL) to compare them with the distributions of the striatal outputs to M1 and mPFC (*Figure 1*, *Figure 1—figure supplement 3C and D*).

For the Rt-RABV tracing, we analyzed the population of starter cells in SNr that project to VA-VL motor thalamus. To determine whether the mCherry+ VA-VL projecting cells were non-dopaminergic SNr neurons, we counted TH+ dopaminergic cells and mCherry+ cells in the midbrain including VTA, SNc and SNr in two cases of six mice, and found that fewer than 4% of neurons were TH-positive. Also, we quantified the percentage of TH+ cells among all the starter cells, and the percentage of nigro-thalamic cells (mCherry+) among all the TH+ SNr cells. mCherry+/Rt-RABV-GFP+ starter SNr neurons were also represented by the digital reconstruction using Neurolucida for visualization. To normalize the distributions of Rt-RABV+ cells in the striatum, we analyzed sections under the 200 μm of the interval and categorized each Rt-RABV+ neuron into VMS, VLS, DMS, DLS or TS. For this analysis, we counted Rt-RABV+ striatal neurons using either the Neurolucida offline software, or under the epi-fluorescent microscope (Axioskop 2, Zeiss). Referring to the standard mouse brain atlas (*Franklin and Paxinos, 2007*), we defined Rt-RABV+ cells in VS when they were located within the nucleus accumbens in the atlas, approximately ranging from the beginning of the striatum to 0.7 mm anterior to bregma. The border of VMS and VLS, or of DMS and DLS, was determined in the middle of these regions. We also defined the beginning of TS at 0.8 mm posterior to the bregma, so that posterior to this level, all of the Rt-RABV+ neurons were categorized into TS. We also analyzed labeling in ventral pallidum (VP), GPe and STN and quantified the normalized distribution of Rt-RABV+ neurons across these three regions. To delineate VP from the surrounding structure, we referred to the standard mouse atlas (*Franklin and Paxinos, 2007*) and performed substance P antibody staining in a few mice as a reference. To analyze the Rt-RABV-GFP+ cells in SNr (*Figure 3* and *Figure 3—figure supplement 1*), we first counted the number of GFP+ cells in SNr for three sections per mouse, located approximately at AP −3.0,–3.2, and −3.4 mm from bregma. Next, we measured the coordinates of the medial and lateral edges of SNr in each section and determined the fraction of neurons located in either medial or lateral SNr (*Figure 3—figure supplement 1A*) or in four sub-divided regions (*Figure 3—figure supplement 1B*). We then averaged the percentage over the three sections in each animal, and calculated mean values across animals (N = 5).

We analyzed injection sites for each viral/CTb tracing by 3D-reconstructions. We used the Neurolucida software and imported a template coronal section from the standard brain atlas, in which contours of the regions of interests such as cerebral cortex, striatum, and SNr were drawn. These collected contours were reconstructed by the software to produce templates. Subsequently, the injection sites were visualized by projecting the observed infected or injected areas to the templates.

## Ex vivo slice electrophysiology

Ten days before the recording experiment, AAV.hsyn.ChR2.eYFP was injected into DLS or VS (0.3 µl, coordinates mentioned above). Five days before recording, a subsequent injection of 1:3 diluted red retrobeads (0.5 µl) was made in motor thalamus (VA-VL). On the day of recording, mice were deeply anesthetized with a ketamine-xylazine mixture and transcardially perfused with ice-cold NMDG cutting solution, saturated with 95% $O_2$/5% $CO_2$, containing (in mM): NMDG 105, HCl 105, KCl 2.5, $NaH_2PO4$ 1.2, $NaHCO_3$ 26, glucose 25, sodium L-ascorbate 5, sodium pyruvate 3, thiourea 2, $MgSO_4$ 10, $CaCl_2$0.5 (300 mOsm/kg, pH = 7.4). Fresh brains were cut into 250-µm-thick coronal slices on a vibratome (Leica VT1000S) through SNr in ice cold, bubbling NMDG cutting solution. Next, slices were recovered for ~15 min at 33˚C in bubbling NMDG cutting solution followed by 45 min in normal ACSF containing (in mM): NaCl 125, KCl 2.5, $NaH_2PO_4$ 1.25, $NaHCO_3$ 25, D-glucose 12.5, $MgCl_2$ 1, $CaCl_2$2 (295 mOsm/kg, pH = 7.4), at 27˚C. After at least one hour of recovery, slices were transferred to a recording chamber perfusing with ACSF at ~2 mL/min bubbled with 95% $O_2$/5% $CO_2$ at 30˚C. Medial and lateral SNr were visually identified under IR-DIC 10X objectives and regions of interest were confirmed by eYFP expression. Whole-cell recordings were performed on neurons labeled with retrobeads under a 40X objective lens. 3~5 MΩ glass pipettes were pulled on a Sutter P-97 puller. Pipettes were filled with internal solution containing (in mM): 115 CsCl, 10 HEPES, 1 EGTA, 20 TEA-Cl, 5 QX-314 (Br- salt), 4 MgATP, 0.3 Na-GTP, and 8 $Na_2$-phosphocreatine (pH 7.3 adjusted with CsOH; 295 mOsm/kg). After break in, cells were held at −70 mV, and ACSF with 10 µM CNQX (Tocris) and 50 µM DL-APV (Sigma-Aldrich) was perfused into the recording chamber to inhibit AMPA-receptor- and NMDA-receptor-mediated excitatory currents, respectively. ~10 min post drug wash in, the paired pulse light stimulation (473 nm, 5~60 mW/mm$^2$, 2.5 ms, 50 ms ISI) generated by a 473 nm blue DPSS laser system (Laserglow Technologies) was delivered through a 200 µm optic fiber (ThorLabs) positioned close to the patched cell (~50–150 µm) at 0.05 Hz to induce IPSC. 100 µM picrotoxin (PTX, Sigma-Aldrich) was applied in addition to previous drugs about 5 min later to verify the recorded $GABA_A$R-mediated inhibitory current. Access or series resistance ranged from 14 to 25 MΩ and was monitored online. Any changes greater than 20% were omitted from the analysis. Voltage-clamp recordings were performed using a Multiclamp 700A (Axon Instruments), filtered at 3 kHz and digitized at 10 kHz. The paired pulse ratio was calculated as the ratio of 2$^{nd}$ to 1$^{st}$ amplitude.

## In vivo electrophysiology

In vivo recording of M1/mPFC neurons during optogenetic stimulation was performed as previously described (*Jin and Costa, 2010*; *Klug et al., 2018*). Briefly, we injected non-floxed version of AAV-ChR2 virus (0.3 µl of AAV9.hsyn.ChR2.eYFP) into DLS/VS of wild-type mice (using the coordinates as mentioned above). Ten days after the viral injection, the mice were lightly anesthetized using isoflurane (4% induction; 0.5–1.5% sustained) and were placed in a stereotactic frame. For electrophysiological recording, we utilized electrode arrays (Innovative Neurophysiology Inc, Durham, NC) of 16 tungsten contacts (2 × 8) that were 35 µm in diameter. Electrodes were spaced 150 µm apart in the same row and 200 µm apart between two rows. The total length of the electrodes was 5 mm. An array was incrementally lowered into M1 (AP, +1.2; ML, 1.4; DV, –0.2 ~ −1.1) and mPFC (AP, +2.1; ML, 0.3; DV, –1.3 ~ −1.6), allowing us to record neurons at multiple depths in each cortical area. Silver grounding wire was attached to skull screws. Neural activity was recorded using the MAP system (Plexon Inc, Dallas, TX). The spike activities were initially online sorted with a build-in algorithm (Plexon Inc, Dallas, TX). Only spikes with stereotypical waveforms that were clearly distinguished from noise and which had relatively high signal-to-noise ratio were tagged and saved for further analysis. After the recording session, the recorded spikes were further isolated into individual units using offline sorting software (Offline Sorter, Plexon Inc, Dallas, TX). Each individual unit

displayed a clear refractory period in the inter-spike interval histogram, with no spikes during the refractory period (larger than 1.3 ms). To stimulate striatal terminals within SNr optogenetically, we placed an optic fiber (200 μm in diameter) in the medial SNr (AP, −3.4; ML, ±1.1; DV, −3.9) for VS terminal stimulation or in the lateral SNr (AP, −3.4; ML, ±1.6; DV, −3.8) for DLS terminal stimulation. For each recording session, blue laser stimulation was delivered through the optic fiber from a 473 nm laser (Laserglow Technologies, Toronto, ON) via a fiber-optic patch cord, and the neuronal responses were simultaneously recorded. The stimulation patterns included 1 s constant light and 5 or 20 Hz (10 ms pulse width, 5 or 20 pulses in 1 s). The inter-stimulation interval was 4 s and each stimulation pattern was repeated for 30 trials. The laser power was adjusted carefully (~3.0–5.0 mW) to drive reliable response. Peri-stimulus time histogram (PSTH) for each neuron was constructed with 1 ms time bins, aligned to the stimulation onset at 0, and smoothed using a build-in Gaussian filter in MATLAB (Mathworks). The latency of responses to optogenetic stimulation was defined as the start of a significant increase in firing rate or decrease in PSTH during the stimulation period. The thresholds for determining the significant firing change were defined as averaged spontaneous firing rate + 3 × SD (standard deviation) for significant increase, and averaged spontaneous firing rate – SD for significant decrease. We categorized all the recorded neurons into positive (excited), negative (inhibited), and non-responded ones, using a defined criterion at the response latency within 5–35 ms. We also calculated the proportion of neurons that significantly changed their firing rates within 5–15 ms, 5–25 ms and 5–45 ms (*Figure 5—figure supplement 1C–E*). The results indicate the same overall tendency in proportion.

To investigate response latencies at each of the downstream nuclei of striatum (SNr, motor thalamus, and M1) within the striato-nigro-thalamo-cortical loop, while activating the D1R-positive spiny projection neurons in striatum, we injected Cre-dependent AAV5-EF1a-DIO-hChR2(H134R)-mCherry virus into striatum of *Drd1a-Cre* mice (GENSAT, EY217) or genetically expressed ChR2 under cre control (D1-cre × Ai32). The recording procedure was the same as described above. For the SNr recording while stimulating D1 neurons, we carried out both the soma stimulation DLS and terminal stimulation in SNr. We combined these data sets and analyzed the latency distribution collectively.

### Statistical analyses

Data were analyzed by GraphPad Prism 7. For the analysis of RABV+ neurons in the striatum from both Wt-RABV and Rt-RABV tracing studies, we performed a two-way analysis of variance (ANOVA) with injection sites and labeled striatal regions as factors. In the statistical analysis of the open-loop component from the Wt-RABV tracing study, we used an unpaired Student's t-test. Comparing the proportion of responded neurons in M1 and mPFC with striato-nigral terminal stimulation, we compared each combination of two groups using a Z-test. $p < 0.05$ was considered significant.

## Acknowledgements

The authors would like to thank Dr James Connor for his technical advice and comments on the manuscript. We thank Cheng Ye for his technical support in histology. This study was supported by the JSPS Institutional Program for Young Researcher Overseas Visits (SA); the JSPS Grant-in-Aid for Young Scientists (A) (SA); the JSPS Grant-in-Aid for Challenging Exploratory Research (SA); the JSPS Grant-in-Aid for JSPS Fellows (SA and MI); the Dutch Ministry of Health, Welfare, and Sports (TR); the CNRS and Aix-Marseille Université through UMR 7289 (PC); the Human Frontier Science Program (JW); the US National Institutes of Health (R01NS083815 (XJ), R01AG047669 (XJ), and K99NS106528 (JBS)); the Rose Hills Foundation (XJ); and a Mcknight Memory and Cognitive Disorders Award (XJ).

## Additional information

### Funding

| Funder | Grant reference number | Author |
| --- | --- | --- |
| Japan Society for the Promotion of Science | Grant-in-Aid for Young Scientists (A) (17H04749) | Sho Aoki |

| | | |
|---|---|---|
| Dutch Ministry of Health, Welfare and Sport | | Tom JH Ruigrok |
| Centre National de la Recherche Scientifique | | Patrice Coulon |
| Human Frontier Science Program | | Jeffery R Wickens |
| National Institutes of Health | R01NS083815 | Xin Jin |
| Rose Hills Foundation | | Xin Jin |
| McKnight Foundation | Mcknight Memory and Cognitive Disorders Award | Xin Jin |
| Japan Society for the Promotion of Science | Grant-in-Aid for Challenging Exploratory Research (16K12973) | Sho Aoki |
| National Institutes of Health | R01AG047669 | Xin Jin |
| National Institutes of Health | K99NS106528 | Jared B Smith |
| Japan Society for the Promotion of Science | Grant-in-Aid for JSPS Fellows (17J11022) | Sho Aoki |
| Japan Society for the Promotion of Science | Institutional Program for Young Researcher Overseas Visits | Sho Aoki |
| Japan Society for the Promotion of Science | Grant-in-Aid for JSPS Fellows (16J05329) | Masakazu Igarashi |

The funders had no role in study design, data collection and interpretation, or the decision to submit the work for publication.

## Author contributions

Sho Aoki, Conceptualization, Resources, Formal analysis, Funding acquisition, Validation, Investigation, Visualization, Writing—original draft, Writing—review and editing; Jared B Smith, Conceptualization, Formal analysis, Funding acquisition, Validation, Investigation, Visualization, Writing—original draft, Writing—review and editing; Hao Li, Formal analysis, Validation, Investigation, Visualization, Methodology, Writing—review and editing; Xunyi Yan, Validation, Investigation, Visualization, Methodology; Masakazu Igarashi, Formal analysis, Investigation; Patrice Coulon, Resources, Funding acquisition, Investigation; Jeffery R Wickens, Conceptualization, Resources, Supervision, Funding acquisition, Writing—review and editing; Tom JH Ruigrok, Conceptualization, Resources, Formal analysis, Supervision, Funding acquisition, Validation, Investigation, Visualization, Methodology, Writing—review and editing; Xin Jin, Conceptualization, Resources, Supervision, Funding acquisition, Validation, Visualization, Methodology, Writing—review and editing

## Author ORCIDs

Sho Aoki https://orcid.org/0000-0002-8160-6543
Jared B Smith https://orcid.org/0000-0002-0273-4898
Patrice Coulon http://orcid.org/0000-0003-2996-405X
Jeffery R Wickens https://orcid.org/0000-0002-8795-1209
Tom JH Ruigrok https://orcid.org/0000-0001-5537-1165
Xin Jin https://orcid.org/0000-0002-1106-4013

## Ethics

Animal experimentation: All procedures related to trans-synaptic wild-type rabies tracing were carried out in accordance with the European guidelines for the care and use of laboratory animals and with the guideline of the French Ministry for Agriculture and Fisheries, Division of animal rights. They were approved by the ethics committee in Neuroscience at the INT (nr. 02167.01). Recombinant monosynaptic rabies experiment, the other tracing experiments, and electrophysiological

experiments were conducted at the Salk Institute for Biological Studies according to NIH guidelines, and all the protocols were approved by the Institutional Animal Care and Use Committee at the institute.

## Decision letter and Author response
Decision letter https://doi.org/10.7554/eLife.49995.020
Author response https://doi.org/10.7554/eLife.49995.021

## Additional files

### Supplementary files
• Transparent reporting form
DOI: https://doi.org/10.7554/eLife.49995.018

### Data availability
Source data have been provided for ex vivo recording in Figure 4 and a source code used for in vivo recording has also been available.

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
