## [Decision Letter]

[Editors’ note: a previous version of this study was rejected after peer review, but the authors submitted for reconsideration. The first decision letter after peer review is shown below.]

Thank you for submitting your work entitled "An open cortico-basal ganglia loop allows limbic control over motor output via the nigrothalamic pathway" for consideration by *eLife*. Your article has been reviewed three reviewers and the evaluation has been overseen by a Reviewing Editor and a Senior Editor. The following individuals involved in review of your submission have agreed to reveal their identity: Naoshige Uchida (Reviewer #2).

Our decision has been reached after consultation between the reviewers. Based on these discussions and the individual reviews below, we regret to inform you that your work will not be considered further for publication in *eLife*.

While the reviewers found the subject of the study to be of strong general interest, and the conclusions potentially compelling, there were a number of potentially serious technical concerns raised. During the consultation process, all involved agreed that the technical concerns need to be fully addressed before publication in *eLife* could be considered. It was also agreed that the new experiments required would be expected to take more than three months, and therefore this paper should be rejected according to *eLife* policy. However, the authors may be able to address most, if not all, concerns by additional control experiments. If all points are addressed, including fair discussions of remaining caveats, this work may become worthy of publication at *eLife*. Below is a summary of the consultation, followed by the individual reviews:

1) The authors should address all of the technical concerns raised by reviewer 1 and reviewer 2. In particular, the possibility of direct labeling and the difficulty of interpreting multi-step rabies labeling should be appreciated and investigated as much as they can. The following priorities were identified, based on the concerns about the specificity of the circuit tracing raised in the individual reviews:

(1) Alternative possibilities (e.g. GPe-> cortex, STN->cortex) should be investigated.

(2) Investigation of shorter survival time to narrow alternative possibilities (though this may still be not complete).

(3) A better experiment (retro-Cre in motor thalamus, inject retro-Cre in M1 and inject G-deleted rabies along with TVA-RG in meotor thalamus).

(4) Latency issue: The firing rate of M1 neurons show a very sharp peak with almost no latency (in particular, Figure 4E, 4I, Figure 5—figure supplement 1E, Figure 5—figure supplement 1F but also Figure 4M, 4Q). This suggests a possible technical flaw (the possibility of antidromically activating cortical projection neurons). It is certainly much faster than what would be predicted, based for example on Oldenberg and Sabatini, 2015 and Freeze et al., 2013. The latencies of the electrophysiological responses should be well-quantified and described within the context of previous results and expected latencies given the inferred circuit architecture.

(5) The authors should consider redoing experiments with D1-Cre line that would minimize leak into cortex.

2) All reviewers agreed with the points raised by reviewer 2 and reviewer 3, that there was insufficient appreciation of the existing literature, and that the contributions of the work need to be better defined within that context.

*Reviewer #1:*

This paper uses a variety of anatomical and functional approaches to examine the topography of circuit loops between cortex, striatum and thalamus. As the title implies, the authors main finding is that the loop though limbic regions of ventral striatum is "open" in that it influences motor regions of thalamus and cortex. What makes it open is that the motor regions of cortex to not reciprocally control limbic regions of striatum. In addition to this finding, there are many details about the topography of cortical-striatum-thalamus loops that are presented.

The main objection to the study is that it has been carried out with a narrow view of the architecture of these loops. For example, potential movement of virus through projections to cortex from STN and globus pallidus are not considered and only minor consideration is given to return thalamic input to striatum. This is especially troubling when considered the rabies transsynaptic tracing in which it is assumed that neurons can be read out as third order because of the amount of time that has passed. Given the large differences in axon lengths in the circuit and the existence of these short-cut pathways to cortex, it is not clear that the circuit assumptions of the authors are valid. Equally problematic, and discussed below, is the finding that SNR can influence cortical firing in a few tens of ms. This suggests to me that the CHR2 virus injection has, either through leak or retrograde trafficking, resulted in expression in cortex and that cortico-SNR axons are being directly stimulated, thus triggered rapid action potentials in cortex.

*Reviewer #2:*

It has been proposed that the cortex, the basal ganglia and the thalamus form multiple, largely-independent loops. The authors examined the nature of connections in the cortico-basal ganglia loops using various tracing methods and optogenetics in mice. The results demonstrate that largely closed loops exit for the primary motor cortex (M1), secondary motor cortex (M2), and medial prefrontal cortex (mPFC). In addition to these "closed" loops, the authors show that M1 receives polysynaptic inputs from wider regions of the striatum than the dorsolateral striatum (DLS) to which M1 mainly projects, suggesting "open" loops. The authors examined the precise connectivity for such an open loop, and show that the ventral striatum (VS) projects to the medial portion of the substantia nigra pars reticulata (SNr), which then projects to M1 via motor thalamus.

This study uses modern circuit tracing tools and fills various "gaps" in the literature regarding the connectivity in cortico-basal ganglia loops. These results are important and provide more precise information regarding the connectivity than the previous work. However, this work does not necessarily change our present view of the global organization of the cortico-basal ganglia circuit, and the current manuscript does not properly appreciate the previous results. I believe that this work still contains important data worthy of publication at *eLife*, but more careful discussions are needed before publication of this work.

Major issues:

1) One of the main findings, as framed in the current manuscript, is that the "results unveil an open cortico-basal ganglia loop whereby limbic information could modulate motor output through ventral striatum (VS) control of M1". The authors further describe that "Despite largely closed loops within each functional domain, we discovered a novel unidirectional influence of the limbic loop onto the motor loop via VS-substania nigra (SNr)-motor thalamus circuitry". However, this connectivity and the presence of open loop connection are not completely novel.

For instance, in the classic paper (Alexander et al., 1986) which emphasized closed loops, the authors first appreciate that "from earlier data it had appeared that the basal ganglia served primarily to integrate diverse inputs from the entire cerebral cortex and to "funnel" these influences, via the ventrolateral thalamus, to the motor cortex (Allen and Tsukahara, 1974, Evarts and Thach, 1969, Kemp and Powell, 1971). In particular, the basal ganglia were thought to provide a route whereby influences from the cortical association areas might be transmitted to the motor cortex and thereby participate in the initiation and control of movement". Throughout this paper, the authors appreciate the data supporting open connections while emphasizing the dominance of closed loops. As cited by the authors, subsequent studies have also observed evidence for open loops. In particular, Miyachi et al., (1996) describe the precisely the projection from the VS to the motor cortex – the main topic of the present study: it describes that "the cortico-basal ganglia motor circuits involving the dorsal putamen and the STN may constitute separate closed loops based on the somatotopy, while the VS provides common multisynaptic projections to all body-part representations in the MI". Thus, the existing literature points to largely closed natures of this circuit while already appreciating open connections.

Most of the results in the present study are within our current understanding of the global structure. In my opinion, the present result does not overturn the previous view, and the novelty of this study lies in the more precise and systematic characterizations of the connectivity. Although these previous works were primarily on monkeys, the novelty of the existence of the connection from the VS to M1 has to be discussed more carefully.

2) In Figure 1, there are a couple of points that are mischaracterized or may warrant more accurate descriptions that are more in line with the data.

2a) Although it is true that anterograde and retrograde labeling largely overlapped for mPFC, the VS is not necessarily the site of overlap. This appears to indicate that at least a large portion of the VS, in particular the ventro-medial part, is also a case for an open loop input for mPFC. Similar to Miyachi et al., (2006), this part of the VS may be the common source of "open loop" projections to different cortical areas.

2b) The authors emphasize the one-direction connection from the VS to M1/M2. However, the M1/M2 receives polysynaptic inputs from almost all regions of the striatum, encompassing the dorsomedial striatum and the tail of the striatum also, consistent with 'funneling'. This point should be more emphasized. To quantitatively demonstrate this point, a similar quantification as Figure 1M for anterograde labeling would be very useful.

3) Monosynaptic rabies tracing requires control labeling with TVA and rabies but without RG, to examine the location of primary neurons because the threshold of fluorescence detection is much higher than the threshold of TVA expression for rabies infection (that is, rabies virus can infect a cell even when TVA is not detected by fluorescence).

*Reviewer #3:*

Aoki and colleagues report a well-designed and described neuroanatomical study that examines circuitry of cortico-striato-thalamo-cortical loops. Prior to this study, there has been well-described connectivity that establishes regional segregation within the striatum based on cortical inputs as well as outgoing striatal connectivity to thalamic subregions. These observations have led to views of parallel functional distinct subcircuits and uncertainty at the neuroanatomical level as to how the circuits may interconnect to influence each other. This study addresses this important question. Using an assortment of cholera toxin and viral genetic tracing approaches, the teams of investigators show compelling evidence for "cross-loop" connectivity of ventral striatum into dorsal striatal cortical input regions. Importantly, they do not find similar evidence for dorsal striatum crossing over to ventral cortical input regions. Experiments in Figure 2 are an elegant and important complement to the observations in Figure 1 using a distinct viral strategy. The tri-color CtB experiments in Figure 3 provides a powerful view into the thalamic anatomy. Finally, functional electrophysiological evidence of circuit connectivity is included to support key observations. Throughout the manuscript, methods, analytical processes, and each injection case are clearly documented. Overall, the attention to detail in representing the anatomical distributions and the expertly designed technical labeling approaches make this study a landmark resource for the field.

1) Results from both Figure 1 and Figure 2 identify the tail of the striatum as one of the most strongly labelled sites. This is an unexpected and potentially important finding. In reference to Figure 2G bottom right panel, can you clarify the relative position of this highly GFP+ structure to the SNr injection site and comment on the possibility that GFP signal represents spread from direct injection site infection and not by retrograde spread?

2) Do the authors have any data regarding M2 for the experiments in Figure 4? As it stands, the summary figure panel U implies that VS goes through M1 and NOT M2, not just that it wasn't evaluated. I also wonder whether the VS contributions may be greater than in M1.

3) The authors somewhat overstate the field's stance/evidence that the loops are parallel and closed. The thalamo-cortical part of the CSTC loops has been a major open question as to how they relate to the originating cortical inputs. This work is a major asset for answering this question.

4) The literature citations are a bit spotty at times. At a minimum, the authors should include Belin and Everitt, 2008 for prior work demonstrating ventral to dorsal basal ganglia circuit connectivity.

[Editors’ note: what now follows is the decision letter after the authors submitted for further consideration.]

Thank you for resubmitting your work entitled "An open cortico-basal ganglia loop allows limbic control over motor output via the nigrothalamic pathway" for further consideration at *eLife*. Your revised article has been favorably evaluated by Catherine Dulac (Senior Editor), a Reviewing Editor, and three reviewers.

The manuscript has been improved but there were some remaining issues that need to be addressed. Please see the specific points raised by reviewer 1 and reviewer 2, below.

*Reviewer #1:*

The authors have performed additional experiments and addressed most of the previous concerns. I have one remaining issue regarding the new control experiments for localizing starter cells (Figure 2—figure supplement 1F, G).

The importance of this control experiment is because TVA is so effective in supporting rabies infection. Even if TVA-mCherry is not detectable in a standard method, it remains possible that neurons are directly infected by rabies virus. This type of infection can occur not only from cell bodies but also from axons. The only way to know the distribution of starter cells is to perform the control experiments without RG. Here the critical question is not only the distribution of starter cells in SNr but, more critically, whether labeled neurons exist elsewhere in the brain, which could be characterized as input neurons if the control results were not taken into account. Please make sure to report whether labeled neurons existed in other parts of the brain. The lack of this control experiments in many existing papers makes it very hard to interpret the results. Please understand the issue and explicitly describe these numbers quantitatively.

*Reviewer #2:*

The authors have revised the manuscript in a way that makes the conclusions more convincing. The experiments with retro-Cre, shorter rabies exposure, and in vivo recordings using D1-Cre mice have strengthen the paper.

It is difficult to compare histology sections of SNr across different experiments, due to the low brightness of the DAPI and arbitrary drawn outline of SNr that seems to differ for every experiment (Figure 3E, Figure 4B, E, and G). The boundaries of what the authors call medial SNr is not specified, and depending on that definition, one can draw various different conclusions. A better comparison across histological sections with predefined boundaries and clearly visible DAPI staining would convince the readers the evidence for an open loop in BG.

*Reviewer #3:*

I like the additional experiments and am supportive of publishing the manuscript. The new experiments further reduce the possibility of a completely erroneous interpretation of their data. But again, the authors are to be commended on including multiple non-overlapping methodologies to investigate the circuitry they describe.

---

## [Author Response]

[Editors’ note: the author responses to the first round of peer review follow.]

Our decision has been reached after consultation between the reviewers. Based on these discussions and the individual reviews below, we regret to inform you that your work will not be considered further for publication in eLife.While the reviewers found the subject of the study to be of strong general interest, and the conclusions potentially compelling, there were a number of potentially serious technical concerns raised. During the consultation process, all involved agreed that the technical concerns need to be fully addressed before publication in eLife could be considered. It was also agreed that the new experiments required would be expected to take more than three months, and therefore this paper should be rejected according to eLife policy. However, the authors may be able to address most, if not all, concerns by additional control experiments. If all points are addressed, including fair discussions of remaining caveats, this work may become worthy of publication at eLife. Below is a summary of the consultation, followed by the individual reviews:1) The authors should address all of the technical concerns raised by reviewer 1 and reviewer 2. In particular, the possibility of direct labeling and the difficulty of interpreting multi-step rabies labeling should be appreciated and investigated as much as they can. The following priorities were identified, based on the concerns about the specificity of the circuit tracing raised in the individual reviews:(1) Alternative possibilities (e.g GPe-> cortex, STN->cortex) should be investigated.(2) Investigation of shorter survival time to narrow alternative possibilities (though this may still be not complete).

We thank you for these excellent points. There is evidence that GPe projects directly to the frontal cortex (Saundars et al., 2015; Chen et al., 2015), and there is also a report that suggests a direct projection from STN to cortex (Jackson and Crossman, 1981). To thoroughly address this concern, we performed three experiments in the revision as follows:

First, as the reviewers suggested, we have attempted to investigate these circuits that might provide an alternative route for wild-type rabies transfection. Therefore, we performed wild-type rabies tracing from M1 using a shorter survival time, and analyzed differences in the pattern of Wt-RABV labeling between the shorter (58 hours) and longer survival times (70 hours). As shown below (updated Figure 1—figure supplement 1), motor thalamus (including both first and second order transfection) shows the densest Wt-RABV+ labeling even with the survival time of only 58 hours, suggesting the motor thalamus represents first order neurons. Similarly, SNr and thalamic reticular nucleus (TRN) have very clear Wt-RABV+ neurons after 58 hours, indicating that SNr and TRN are second order neurons. In stark contrast, we found almost no labeled neurons in GPe, STN and striatum after 58 hours of survival time, which revealed that GPe, STN and striatum are unlikely to be first or second order neurons. However, both GPe, STN and striatum showed an abundance of Wt-RABV+ cells in the longer 70 hour survival time, indicating that they are most likely third order neurons to the cortical rabies injections. These results suggest two major implications: (1) GPe and STN are unlikely to be a major part of first or second order neurons, and (2) striatal labeling in the longer survival time was largely mediated by SNr and motor thalamus, but not via GPe and STN. This new evidence from the shorter survival time strongly indicates that the vast majority if not all of labeled neurons in the striatum (Figure 1) are likely mediated through the canonical striato-nigro-thalamo-cortical pathway.

Second, as the reviewer pointed out, the use of the shorter survival time might not completely exclude the possible involvement of the GPe → cortex circuit. Thus, we have attempted monosynaptic rabies tracing from pallido-cortical neurons (see following panel). In this experiment, we made the AAVretro.Cre injection in M1 and TVA.RG in GPe. After three weeks, we injected Gdeleted rabies in GPe and tried to identify both starter populations in GPe, and secondary input neurons to GPe (panel A). However, we failed to find any rabies GFP+ / TVA.mCherry+ cells in GPe (Panel B). Even closer examination of TVA-mCherry expression turned out to be negative as well. Indeed, available evidence regarding GPe projections to frontal cortex appears to target the most rostral regions of frontal cortex (Figure 1 of Saunders et al., 2015), and Fr2 (the rostral M2 area, Chen et al., 2015). Taken together, the close observation of the previous studies, this additional experiment showing an absence of primary infections of TVA and rabies GFP in GPe following M1 injection (following panel), and the weak labeling of Wt-RABV+ cells in GPe with a shorter survival time shown above (Figure 1—figure supplement 1), it seems very unlikely that the GPe → cortex circuit was involved in the results of our Wt-RABV tracing.

Third, in order to determine whether and how strongly STN projects to the cortex, we performed Cre-dependent anterograde tracing using Pitx2-Cre mice in which Cre expression is fairly selective to the STN (Schweizer et al., 2014). Author response image 1 shows projections from STN (panel A, B) in Pitx2-Cre mice. As expected, we found dense axonal projections in GPe (panel C) and SNr (panel D), but no axon terminals were visible in the cortex, including M1 (panel E). Additionally, as shown in Figure 1—figure supplement 1, we observed almost no Wt-RABV labeling in STN with the shorter survival time. Therefore, it is unlikely that STN projections to cortex are a major confound in interpreting striatal labeling from WT-RABV cortical injections.

Based on these additional experiments and analyses, we are confident that neither GPe- nor STNcortical projections are the major sources that mediate the results of wild-type rabies transfection in striatum. Instead, the nigro-thalamic pathway seems to mostly mediate this labeling (Figure 1—figure supplement 1). We have now added these additional data to the results and further clarified the findings in subsection “subsection “Trans-synaptic tracing using wild-type rabies virus reveals both closed and open cortico basal ganglia-thalamocortical loops”.

Please note that an admitted caveat of using wild-type rabies for circuitry tracing is the possible involvement of additional pathways. That’s why we went on to validate these pathways with additional monosynaptic tracing experiments and electrophysiological recording in the paper. Nevertheless, as the reviewers suggested, we have added a statement in the revision about the limitation and interpretation of wild-type rabies tracing results, in which we have now stated (subsection “Trans-synaptic tracing using wild-type rabies virus reveals both closed and open cortico basal ganglia-thalamocortical loops”):

“This finding allows us to infer that the striatal labeling in the following experiments are largely mediated by the canonical striato-nigro-thalamo-cortical pathway (Figure 1A), although we cannot completely rule out the possible minor contribution of direct GPe and STN inputs to cortex with this methodology”.

(3) A better experiment (retro-Cre in motor thalamus, inject retro-Cre in M1 and inject G-deleted rabies along with TVA-RG in motor thalamus).

We thank the reviewers for this excellent suggestion. In this revision, we have added this suggested experiment in Figure 3. Accordingly, we injected AAVretro.Cre in M1 and TVA.RG virus in motor thalamus, followed by G-deleted rabies virus in the motor thalamus after 3 weeks (Figure 3A). The starter cells were identified in the motor thalamus (Figure 3B). Also, trans-synaptically labeled GFP+ cortico-thalamic cells were selectively located in M1 (Figure 3C) and GFP+ cerebello-thalamic neurons were identified in dentate and interpositus nuclei (Figure 3D), which indicates that we successfully traced from the motor thalamus. Most critically, we found GFP+ cells in both medial and lateral SNr (Figure 3E) with the distribution of approximately 40% in medial and 60% in lateral SNr (Figure 3F, N = 5). This finding indicates that both limbic/associative medial SNr area as well as the sensorimotor lateral region of SNr possess synaptic inputs onto motor thalamus, which then innervates M1. We believe that this evidence greatly supports our claim about limbic-to-motor connectivity (medial SNr to motor thalamus, Figure 3G). This result also suggests “funneling” from basal ganglia output of all functional domains to motor cortex, given the fact that virtually all of SNr subregions have di-synaptic connections to M1 via motor thalamus. Accordingly, we added a new Results section describing this additional experiment.

(4) Latency issue: The firing rate of M1 neurons show a very sharp peak with almost no latency (in particular, Figure 4E, 4I, Figure 5—figure supplement 1E, Figure 5—figure supplement 1F but also Figure 4M, 4Q). This suggests a possible technical flaw (the possibility of antidromically activating cortical projection neurons). It is certainly much faster than what would be predicted, based for example on Oldenberg and Sabatini, 2015 and Freeze et al., 2013. The latencies of the electrophysiological responses should be well-quantified and described within the context of previous results and expected latencies given the inferred circuit architecture.

We have further researched the literature and the papers mentioned by the reviewer regarding the response latency across the cortico-striato-nigro-thalamocortical loops. The relevant studies are listed below, including the information about their latency results.

Striato-nigral pathway (Striatum → SNr):

1) Freeze et al., (2013), Figure 2D; Optogenetic activation of the direct pathway neurons by stimulation in striatum inhibits SNr neurons at a median latency of 20 ms.

Nigro-thalamic pathway (SNr → thalamus):

2) Tanibuchi et al., (2009b), Figure 2C; Electrical activation of SNr inhibits thalamic neurons at a latency of 2.4 ms.

3) Tanibuchi et al., (2009a), Figure 6C; Electrical activation of SNr inhibits VA-VL thalamic neurons at a mean latency of 2.3 ms.

Striato-nigro-thalamic pathway (Striatum → SNr → Motor thalamus):

4) Lalive et al., (2018), Figure 6d; Optogenetic activation of the direct pathway neurons in dorsal striatum excites most thalamic neurons at a latency of 10-20 ms.

Striato-nigro-thalamic–cortical pathway (Striatum → SNr → Motor thalamus → M1):

5) Oldenburg and Sabatini, (2015), Figure 3A; Direct pathway optogenetic activation changes M1 responses at a mean latency of 123 ms.

Therefore, given the quick transmission from striatum to thalamus (< 10-20ms) and the monosynaptic connection from motor thalamus to cortex, it is plausible to alter cortical activity by activating striatum at latencies much shorter than 100 ms. Oldenburg and Sabatini, (2015) focused on overall effects on the M1 neuronal activity when the striatal direct or indirect pathway is activated, so that they collected the *activity of any M1 neurons, at any layers*, responding to striatal activation and calculated the response latency as an average. By contrast, the aim of the present study is to physiologically identify the likelihood of direct pathway connectivity between striatum and cortex for each possible interaction (i.e. DLS → M1, VS → M1, DLS → mPFC and VS → mPFC). Thus, we have restricted our analysis to the shortest responses that are likely mediated by only the direct striato-nigro-thalamo-cortical pathway (Figure 5). This specific purpose requires our analysis to minimize the number of synapses involved and exclude any possible network effects either from the M1 local circuitry or beyond the striato-nigro-thalamo-cortical pathway. We have thus made a rather strict criterion for the detection of responsive neurons at a latency of <40 ms after the onset of optogenetic stimulation, in order to avoid any false positives.

To further verify the latency of cortical responses to striatal activation, we investigated response latencies at each of the downstream nuclei of striatum (SNr, motor thalamus, and M1), while activating the D1R-positive spiny projection neurons in striatum. To optogenetically activate D1 spiny projection neurons, we injected Cre-dependent AAV-ChR2 virus in striatum of D1-Cre mice or genetically expressed ChR2 under cre control (D1-Ai32). These data are included in Figure 5—figure supplement 2. First, the inhibition of SNr neurons occurs at a latency of 3-10 ms after the onset of optical stimulation (Figure 5—figure supplement 2B, C, D). We next found the shortest latency of the motor thalamus responses to D1 striatal activation was at 5-10 ms (Figure 5—figure supplement 2G, H, I), which is slightly shorter than a previous study (Lalive et al., 2018). Finally, a group of M1 neurons responded to striatal D1 activation within a range of 9-35 ms (Figure 5—figure supplement 2L, M, N). In sum, these results indicate that the activation of striatal D1R spiny projection neurons can alter cortical activity at a much shorter latency (9-35 ms) compared to the average latency reported by Oldenburg and Sabatini, which supports the latency data from our experiments activating striatal terminals in SNr and recording from cortex (Figure 5).

Regarding the potential antidromic response caused by any leakage of the virus into the cortex, we have no evidence that non Cre-dependent virus injected into the striatum was leaked into the cortex, based on histological analysis. Additionally, the injection and recording coordinates along anteriorposterior and medio-lateral axes were far from each other. Yet, given the existence of cortico-nigral projections (Naito and Kita, 1994) as the reviewer pointed out, we have now added a lower limit to the criterion of the latency in the revision to exclude any response less than 5 ms based on published data (Li et al., 2015).

Antidromic response of pyramidal tract neurons in the motor cortex (antidromic, M1 → Pons):

(6) Li et al., (2015), Extended Data Figure 5B and 5E; Activation of axon terminals of M1 pyramidal neurons at pons yields cortical excitation at 2.7 ms (pyramidal tract neurons).

With this criterion, our conclusion is still consistent that there is one-way connectivity from VS to M1, in addition to predominant within-domain connectivity (DLS to M1 and VS to mPFC) (Figure 5T; Figure 5—figure supplement 1C-E), with almost no influence of DLS on mPFC. In this revision, we have also included the data analyses for the fraction of responsive neurons using four different criteria, in which we defined cortical responsive neurons in varying time windows up to 45 ms beginning from 5 ms after the onset of the optogenetic stimulation (5-15 ms, 5-25 ms, 5-35 ms (used for Figure 5), and 5-45 ms). The results based on these four different criteria indicate the same quantitative distributions and consistent statistical results, as shown in Figure 5 and Figure 5—figure supplement 1.

In summary, in response to the reviewer’s comments, we have re-analyzed electrophysiological data in a more strict and conservative way to define cortical responses to the activation of the striatonigro-thalamo-cortical pathway. Our conclusion that there is a one-way limbic to motor connectivity remains unchanged. Accordingly, the revision has now included: (1) updated Figure 5, and updated Figure 5—figure 18, (2) new electrophysiological recording at each step downstream from striatum (new Figure 5—figure supplement 2) with statistics of the fraction of responsive neurons, and (3) additional literature supporting our latency data as well as statements clarifying how we define responsive neurons using our new criteria. Accordingly, we have made a paragraph for the clarification in the revision (subsection “in vivooptogenetic stimulation confirms that ventral striatum controls motor cortex”).

(5) The authors should consider redoing experiments with D1-Cre line that would minimize leak into cortex.

As we mentioned above, the injection sites of non-Cre-dependent ChR2 in DLS and VS do not match the cortical recording sites, so it is unlikely that the leakage of the ChR2 virus into the cortex causes antidromic responses in the cortex. Nonetheless, as shown above, we have now added the experiments suggested by the reviewers using D1-Cre mice (GENSAT, EY217), and also performed the optogenetic stimulation of D1R-positive spiny projection neurons in the VS and recorded M1 to replicate our original results. Consistent with our original finding (Figure 5), this experiment also showed that 8% of recorded neurons in M1 responded to the optical stimulation within 5-35 ms (Figure 5—figure supplement 2P-T). Our new data showing that the optogenetic stimulation of D1R VS neurons alters M1 activity supports our conclusion. We have added a result for this experiment to the main text (subsection “in vivooptogenetic stimulation confirms that ventral striatum controls motor cortex”).

2) All reviewers agreed with the points raised by reviewer 2 and reviewer 3, that there was insufficient appreciation of the existing literature, and that the contributions of the work need to be better defined within that context.

We appreciate this comment and have reviewed the suggested literature by those referees. We have searched for other previous studies on the topic of “open loops” in basal ganglia circuits, including experimental and conceptual work, all of which have now been included in the revised manuscript. Accordingly, we have modified the manuscript to discuss the previous work and clarified our language to state that our major contribution is on (1) more concretely demonstrating the existence of open-loops with modern genetic and viral techniques, (2) demonstrating the precise mechanism by which the limbic-to-motor interaction occurs at the medial SNr to motor thalamus projection, and (3) demonstrated the functional aspect of these anatomical connections with both in vitro and in vivo electrophysiology. We believe that this revision fits better into the current framework of the fields’ view of cortico-basal ganglia-thalamocortical loops and further states our major contribution to clarifying details of the open-loop component. These citations and clarifications are now added into both the Introduction, Results section and Discussion section.

Reviewer #1:This paper uses a variety of anatomical and functional approaches to examine the topography of circuit loops between cortex, striatum and thalamus. As the title implies, the authors main finding is that the loop though limbic regions of ventral striatum is "open" in that it influences motor regions of thalamus and cortex. What makes it open is that the motor regions of cortex to not reciprocally control limbic regions of striatum. In addition to this finding, there are many details about the topography of cortical-striatum-thalamus loops that are presented.The main objection to the study is that it has been carried out with a narrow view of the architecture of these loops. For example, potential movement of virus through projections to cortex from STN and globus pallidus are not considered and only minor consideration is given to return thalamic input to striatum. This is especially troubling when considered the rabies transsynaptic tracing in which it is assumed that neurons can be read out as third order because of the amount of time that has passed. Given the large differences in axon lengths in the circuit and the existence of these short-cut pathways to cortex, it is not clear that the circuit assumptions of the authors are valid.

As suggested, we have attempted to investigate GPe and STN projections to cortex as alternative circuits that might provide a route for wild-type rabies transfection. To this end, we performed wildtype rabies tracing from M1 using a shorter survival time, and analyzed differences in the pattern of Wt-RABV labeling between the shorter (58 hours) and longer survival times (70 hours). As shown below in the updated Figure 1—figure supplement 1, motor thalamus (including both first and second order transfection) shows the densest Wt-RABV+ labeling even with the survival time of only 58 hours, suggesting the motor thalamus represents first order neurons. Similarly, SNr and thalamic reticular nucleus (TRN) have very clear Wt-RABV+ neurons after 58 hours, indicating that SNr and TRN are second order neurons. In stark contrast, we found almost no labeled neurons in GPe, STN and striatum after 58 hours of survival time, which revealed that GPe, STN and striatum are unlikely to be first or second order neurons. However, both GPe, STN and striatum showed an abundance of Wt-RABV+ cells in the longer 70 hours survival time, indicating that they are most likely third order neurons to the cortical rabies injections. These results suggest two major implications: (1) GPe and STN are unlikely to be a major part of First or second order neurons, and (2) striatal labeling in the longer survival time was largely mediated by SNr and motor thalamus, but not via GPe and STN. This new evidence from the shorter survival time strongly indicates that the vast majority if not all of labeled neurons in the striatum (Figure 1) are likely mediated through the canonical striato-nigro-thalamo-cortical pathway.

Please note that an admitted caveat of using wild-type rabies for circuitry tracing is the possible involvement of additional pathway, and that is why we went on to validate these pathways by following additional monosynaptic tracing experiments and electrophysiological recording in the paper. Nevertheless, as the reviewers suggested, we have added a statement in the revision about the limitation and interpretation of wild-type rabies tracing results, in which we have now stated: (subsection “Trans-synaptic tracing using wild-type rabies virus reveals both closed and open cortico basal ganglia-thalamocortical loops”).

“This finding allows us to infer that the striatal labeling in the following experiments are largely mediated by the canonical striato-nigro-thalamo-cortical pathway (Figure 1A), although we cannot completely rule out the possible minor contribution of direct GPe and STN inputs to cortex with this methodology”.

To determine the survival time that allows third order labeling but not fourth order, our previous study has thoroughly investigated and adjusted our survival time to 70 hours, so as to avoid fourth order transfection. When we injected wild-type rabies into M1, we found clear labeling in the ipsilateral striatum and STN after 70 hours of survival time, but not in the contralateral striatum and STN.

Similarly, when rabies was injected into primary somatosensory cortex (S1), we identified densely labeled cells in the contralateral dorsal root ganglion in the spinal cord without any infection in the ipsilateral side. All of these experiments were performed under the same survival time as the present study. If, as this reviewer speculated, the fourth order labeling had occurred, we should see obvious rabies-infected cells in the contralateral striatum and STN in the case of M1 injection, and those in the ipsilateral dorsal root ganglion when injected in S1. Therefore, we consider that our survival time is not long enough for fourth order infection, and that a possibility that fourth order transfection might cause any confound is very low.

Equally problematic, and discussed below, is the finding that SNR can influence cortical firing in a few tens of ms. This suggests to me that the CHR2 virus injection has, either through leak or retrograde trafficking, resulted in expression in cortex and that cortico-SNR axons are being directly stimulated, thus triggered rapid action potentials in cortex.

As for the potential antidromic response caused by any leakage of the virus into the cortex, we have no evidence that non Cre-dependent virus injected into the striatum was leaked into the cortex, based on histological analysis. Furthermore, injection and recording coordinates along anteriorposterior and medio-lateral axes were far from each other. Yet, given the existence of cortico-nigral projections (Naito and Kita, 1994) as the reviewer points out, we now added a lower limit to the criterion of the latency in the revision to exclude any response less than 5 ms based on published data (Li et al., 2015).

As we mentioned earlier, we have re-analyzed electrophysiological data in a more strict and conservative way to define cortical responses to the activation of the striato-nigro-thalamo-cortical pathway. Our conclusion that there is a one-way limbic to motor connectivity remains unchanged. Accordingly, the revision has now included: (1) updated Figure 5T, Figure 5—figure supplement 1, (2) new electrophysiology recordings at each step downstream from striatum (Figure 5—figure supplement 2) with statistics of the fraction of responsive neurons, and (3) additional literature supporting our latency data as well as statements clarifying how we define responsive neurons using our new criteria. Accordingly, we have made a paragraph for the clarification of our choice in the latency in the revision (subsection “in vivooptogenetic stimulation confirms that ventral striatum controls motor cortex”).

Reviewer #2:[…] 1) One of the main findings, as framed in the current manuscript, is that the "results unveil an open cortico-basal ganglia loop whereby limbic information could modulate motor output through ventral striatum (VS) control of M1". The authors further describe that "Despite largely closed loops within each functional domain, we discovered a novel unidirectional influence of the limbic loop onto the motor loop via VS-substania nigra (SNr)-motor thalamus circuitry". However, this connectivity and the presence of open loop connection are not completely novel.For instance, in the classic paper (Alexander et al., 1986) which emphasized closed loops, the authors first appreciate that "from earlier data it had appeared that the basal ganglia served primarily to integrate diverse inputs from the entire cerebral cortex and to "funnel" these influences, via the ventrolateral thalamus, to the motor cortex (Allen and Tsukahara, 1974, Evarts and Thach, 1969, Kemp and Powell, 1971). In particular, the basal ganglia were thought to provide a route whereby influences from the cortical association areas might be transmitted to the motor cortex and thereby participate in the initiation and control of movement". Throughout this paper, the authors appreciate the data supporting open connections while emphasizing the dominance of closed loops. As cited by the authors, subsequent studies have also observed evidence for open loops. In particular, Miyachi et al., (1996) describe the precisely the projection from the VS to the motor cortex – the main topic of the present study: it describes that "the cortico-basal ganglia motor circuits involving the dorsal putamen and the STN may constitute separate closed loops based on the somatotopy, while the VS provides common multisynaptic projections to all body-part representations in the MI". Thus, the existing literature points to largely closed natures of this circuit while already appreciating open connections.Most of the results in the present study are within our current understanding of the global structure. In my opinion, the present result does not overturn the previous view, and the novelty of this study lies in the more precise and systematic characterizations of the connectivity. Although these previous works were primarily on monkeys, the novelty of the existence of the connection from the VS to M1 has to be discussed more carefully.

Thank you for bringing this concern to our attention. As we responded earlier, we have gone through all the relevant literature and now included these papers in the revision. We have covered experimental studies as well as theoretical studies that have demonstrated or predicted the existence of an open loop structure across different cortico-basal ganglia thalamocortical loops. These citations and clarifications are now added into both the Introduction, Results section and Discussion section part of the manuscript.

2) In Figure 1, there are a couple of points that are mischaracterized or may warrant more accurate descriptions that are more in line with the data.2a) Although it is true that anterograde and retrograde labeling largely overlapped for mPFC, the VS is not necessarily the site of overlap. This appears to indicate that at least a large portion of the VS, in particular the ventro-medial part, is also a case for an open loop input for mPFC. Similar to Miyachi et al., (2006), this part of the VS may be the common source of "open loop" projections to different cortical areas.2b) The authors emphasize the one-direction connection from the VS to M1/M2. However, the M1/M2 receives polysynaptic inputs from almost all regions of the striatum, encompassing the dorsomedial striatum and the tail of the striatum also, consistent with 'funneling'. This point should be more emphasized. To quantitatively demonstrate this point, a similar quantification as Figure 1M for anterograde labeling would be very useful.

We have analyzed the distribution of cortico-striatal terminals from mPFC and M1 across the same five striatal subregions (VMS, VLS, DMS, DLS and TS), the results of which are consistent with previous literature (Voorn et al., 2004; Hintiryan et al., 2016). In the revision, we have compared these data with the distributions of striatal output neurons to the cortex (Figure 1—figure supplement 3). Importantly, as this reviewer suggested, there is a mismatch between input and output in ventro-medial striatum (VMS) as in Figure 1—figure supplement 3C. Likewise, Figure 1—figure supplement 3D shows the exclusive projection from M1 to DLS but rather diverse output sources from striatum to M1, which implies the funnel through striatal outputs to M1.

Accordingly, we have added a new paragraph in the Results section describing these data, and stated that ventromedial part of striatum may be a common source of the open-loop connection (Miyachi et al., 2006): (subsection “Trans-synaptic tracing using wild-type rabies virus reveals both closed and open cortico basal ganglia-thalamocortical loops”third).

“Particularly, VMS had a mismatch in which it receives a few inputs but has many outputs (Figure 2—figure supplement 1C), suggesting a possibility that VMS provides a common source for the open loop structure (Miyachi et al., 2006).”

Also, we have added a statement about the “funneling” (subsection “Trans-synaptic tracing using wild-type rabies virus reveals both closed and open cortico basal ganglia-thalamocortical loops”):

“This result indicates that only DLS receives M1 inputs but multiple striatal regions connect to M1, implying that all striatal outputs are funneled into M1 via basal ganglia output.”

3) Monosynaptic rabies tracing requires control labeling with TVA and rabies but without RG, to examine the location of primary neurons because the threshold of fluorescence detection is much higher than the threshold of TVA expression for rabies infection (that is, rabies virus can infect a cell even when TVA is not detected by fluorescence).

In the original manuscript, we carefully analyzed both TVA.mCherry+ as well as GFP+ cells to define starter cells. We believe that this method is still valid for demonstration of starter populations (Callaway and Luo, 2015), so that we have left the original analysis for the starter cells in the SNr in Figure 2. We appreciate this suggestion and now show the distribution of the starter cells in an alternative way. We have added the suggested control experiment without injecting rabies glycoprotein in Figure 2—figure supplement 1F-G. We found widely distributed GFP+ cells, supporting our original analysis for the starter cells that there is convergence from medial and lateral SNr to motor thalamus, as also revealed by our new Figure 3. Furthermore, we added another case to indicate the starter cells (Figure 2—figure supplement 1A-B) and detailed analyses to characterize their relationship with TH^+^ dopamine cells in response to this reviewer’s minor comment (Figure 2—figure supplement 1C-E).

Reviewer #3:[…] 1) Results from both Figure 1 and Figure 2 identify the tail of the striatum as one of the most strongly labelled sites. This is an unexpected and potentially important finding. In reference to Figure 2G bottom right panel, can you clarify the relative position of this highly GFP+ structure to the SNr injection site and comment on the possibility that GFP signal represents spread from direct injection site infection and not by retrograde spread?

There are no mCherry+ cells in this GFP+ zone located in the tail of the striatum, and indeed injection coordinates into SNr (from bregma, AP -3.3 mm) is rather different from the tail of the striatum (AP -1.8 mm) (i.e., 1.5 mm separation). Thus, the labeling in TS is the result of transsynaptic transport, and not of injection spread. Also, we injected AAVretro.Cre into the motor thalamus. Therefore, given no striatal projections to the thalamus, the tail of striatum cannot be a primary population (starter cells). To avoid confusion, however, we decided to clarify it with a note in the legend that this dense GFP+ population is not starter cells.

Accordingly, we have now stated (Figure 2 legend):

“Note that densely labeled GFP+ cells in TS are not starter cells as there are no TVA.mCherry+ cells.”

2) Do the authors have any data regarding M2 for the experiments in Figure 4? As it stands, the summary figure panel U implies that VS goes through M1 and NOT M2, not just that it wasn't evaluated. I also wonder whether the VS contributions may be greater than in M1.

Thank you for clarifying this potential confusion to the audience. Since we did not test the contribution of VS to M2, we decided not to label M2 in the panel U in the original manuscript (original Figure 4). However, having discussion among the authors, we have now included “M2” next to M1 (Figure 5U in the revision), based on our anatomical experiments. We have treated this Figure 5U as a conclusive remark, so that it is understandable to the audience, and convincing to this reviewer why we put M2 label here.

Indeed, we have put a statement about the overall picture of our finding using this panel U in Discussion section:

“While our data do demonstrate mostly closed cortico-basal ganglia loops within each domain, our results confirm an open cortico-basal ganglia loop allowing a one-way interaction from limbic to motor circuitry (Figure 5U).”

As our focus in this particular experiment (Figure 5) was to investigate physiological connectivity between ventral striatum to M1, we focused on the effect of VS activation onto M1. Yet, it is quite possible that the contribution of VS to M2 is higher than M1. This would be a very good question in the future study asking whether or not there is a gradient in the contribution of ventral striatal output to various motor-related cortical areas.

3) The authors somewhat overstate the field's stance/evidence that the loops are parallel and closed. The thalamo-cortical part of the CSTC loops has been a major open question as to how they relate to the originating cortical inputs. This work is a major asset for answering this question.

Thank you for this comment. We have amended our standpoint about the organization of corticobasal ganglia loops. Accordingly, we have modified the manuscript to discuss the previous work and clarified our language to state that our major contribution is on (1) more concretely demonstrating the existence of open-loops with modern genetic and viral techniques, (2) demonstrating the precise mechanism by which the limbic-to-motor interaction occurs at the medial SNr to motor thalamus projection, and (3) demonstrated the functional aspect of these anatomical connections with both in vitro and in vivo electrophysiology. We believe that this revision fits better into the current framework of the fields’ view of cortico-basal ganglia-thalamocortical loops and further states our major contribution to clarifying details of the open-loop component.

Accordingly, we have cited relevant previous studies in the Introduction:

“Cortico-basal ganglia-thalamocortical loops have been largely conceptualized as closed, functionally segregated loops, in which limbic, associative, and sensorimotor information are processed in parallel (Alexander et al., 1986; Deniau et al., 1996; Haber, 2003; Kim and Hikosaka, 2015; Montaron et al., 1996; Parent and Hazrati, 1995). Alternatively, older studies proposed a “funnel-like” architecture to basal ganglia output, such that all functional loops provide some input to the motor circuit (Allen and Tsukahara, 1974; Kemp et al., 1971). While a “partially-open” loop architecture in the cortico-basal ganglia circuitry has been suggested from primate studies (Joel and Weiner, 1994; Kelly and Strick, 2004; Miyachi et al., 2006), this previous evidence is incomplete and the precise anatomical basis underlying connections between functionally distinct loops has not been identified.”

and Discussion section.

“[…] Cortico-basal ganglia circuitry has been considered to mostly consist of parallel, segregated loops within different functional domains (Alexander et al., 1986; Haber, 2003; Kelly and Strick, 2004; Kim and Hikosaka, 2015; Miyachi et al., 2006; Parent and Hazrati, 1995), with a possibility of some openloop architecture providing interactions between domains (Haber, 2003; Joel and Weiner, 1994; Kelly and Strick, 2004; Miyachi et al., 2006). While our data do demonstrate mostly closed corticobasal ganglia loops within each domain, our results confirm an open cortico-basal ganglia loop allowing a one-way interaction from limbic to motor circuitry (Figure 5U). The open-loop structure we revealed here is consistent with earlier studies in primates that have identified connectivity from ventral putamen to M1 (Kelly and Strick, 2004; Miyachi et al., 2006), with conceptual work on the convergence of basal ganglia outputs to motor circuits (Allen and Tsukahara, 1974; Haber, 2003; Joel and Weiner, 1994; Kemp et al., 1971), as well as with behavioral findings that suggest the involvement of VS in modifying motor output (Belin and Everitt, 2008; Floresco, 2015; Sawada et al., 2015).”

4) The literature citations are a bit spotty at times. At a minimum, the authors should include Belin and Everitt, 2008 for prior work demonstrating ventral to dorsal basal ganglia circuit connectivity.

As we mentioned in the essential revisions above, we have now reviewed and included relevant studies including experimental and conceptual work on the cortico-basal ganglia circuits in the revision, including Belin and Everitt, (2008). Accordingly, we have updated the citations in the Introduction and Discussion section).

[Editors' note: the author responses to the re-review follow.]

The manuscript has been improved but there were some remaining issues that need to be addressed. Please see the specific points raised by reviewer 1 and reviewer 2, below.Reviewer #1:The authors have performed additional experiments and addressed most of the previous concerns. I have one remaining issue regarding the new control experiments for localizing starter cells (Figure 2—figure supplement 1F, G).The importance of this control experiment is because TVA is so effective in supporting rabies infection. Even if TVA-mCherry is not detectable in a standard method, it remains possible that neurons are directly infected by rabies virus. This type of infection can occur not only from cell bodies but also from axons. The only way to know the distribution of starter cells is to perform the control experiments without RG. Here the critical question is not only the distribution of starter cells in SNr but, more critically, whether labeled neurons exist elsewhere in the brain, which could be characterized as input neurons if the control results were not taken into account. Please make sure to report whether labeled neurons existed in other parts of the brain. The lack of this control experiments in many existing papers makes it very hard to interpret the results. Please understand the issue and explicitly describe these numbers quantitatively.

We thank the reviewer for this final point and believe we understand the issue the reviewer is concerned about. The goal of our recombinant, circuit-specific rabies tracing of the nigro-motor thalamic cells was to identify the projection neurons from striatum to SNr (Figure 2). To further specify the starter population in SNr, we performed the control experiment without rabies glycoprotein (RG).

In this revision, we investigated the rabies-labeled neurons in other brain regions for the actual monosynaptic rabies tracing experiment (AAVretro.Cre in motor thalamus, and TVA/RG virus and dG-Rabies virus in SNr, as in Figure 2) as well as the control experiment where the glycoprotein was withheld. Regarding the actual rabies tracing, we found trans-synaptically labeled rabies+ cells in GPe and STN (updated Figure 2—figure supplement 2C and 2D), both of which are known to project to SNr. As per the reviewer’s comment, the quantified data have now been included in Figure 2—figure supplement 2D in addition to the analysis for ventral pallidum and the main text has been updated (subsection “Monosynaptic modified rabies tracing confirms the limbic-to-motor connectivity via the striato-nigro-thalamic pathway”).

As per the reviewer’s comments, the primary concern here pertains to axons/terminals of GPe, and STN taking up the rabies virus directly, and thus might be part of the starter population. However, we are confident that this did not occur in our rabies tracing experiment for several reasons listed below: (1) GPe and STN do not project to motor thalamus, so they cannot express Cre-recombinase through the injection of AAV.retro.Cre into motor thalamus, meaning there is no chance to express rabies glycoprotein (RG) which is necessary for rabies trans-synaptic infection. Thus, it is unlikely for GPe/STN to be the starter population for rabies tracing. (2) STN does not receive striatal inputs and cannot be the starter population that causes striatal labeling. (3) We observed no rabies+ cells in GPe and STN in the control experiment without RG (updated Figure 2—figure supplement 1H), suggesting that direct infection of rabies did not occur in these two regions, and the labeled cells in GPe and STN in the actual rabies tracing experiment are trans-synaptically labeled “input neurons” to the nigro-thalamic cells.

As the goal of the present experiment is to identify the striatal input neurons to nigro-thalamic cells, we further analyzed striatum in the control experiment to completely rule out a possible confound of the direct infection of rabies virus through striato-nigral axons. As shown below, we found no rabies+ striatal cells (updated Figure 2—figure supplement 1H), indicating that the labeling pattern in Figure 2 solely reflects striatal input neurons to nigro-thalamic cells in SNr. For this control experiment, we also analyzed entopeduncular nucleus (EPN) and found no rabies+ cells, suggesting EPN did not contain starter cells either. We agree with the reviewer that demonstrating the absence of starter cells in those nuclei in the control experiment is necessary for readers to interpret the data. As such, we have provided images of striatum, EPN, GPe and STN in Figure 2—figure supplement 1, showing no rabies+ cells in these areas without the use of RG.

We thank the reviewer again for the insightful suggestions. We hope that these new data and control experiments now convincingly demonstrate the validity of the recombinant rabies tracing results.

Reviewer #2:

*The authors have revised the manuscript in a way that makes the conclusions more convincing. The experiments with retro-Cre, shorter rabies exposure, and* in vivo *recordings using D1-Cre mice have strengthen the paper.*

It is difficult to compare histology sections of SNr across different experiments, due to the low brightness of the DAPI and arbitrary drawn outline of SNr that seems to differ for every experiment (Figure 3E, Figure 4B, E, and G). The boundaries of what the authors call medial SNr is not specified, and depending on that definition, one can draw various different conclusions. A better comparison across histological sections with predefined boundaries and clearly visible DAPI staining would convince the readers the evidence for an open loop in BG.

Thank you for bringing up this issue. In response to this comment, we have now added DAPI images or intensified the brightness of DAPI in the figures to ensure the visibility of the spatial extent of SNr. Anatomically, SNr is a region of midbrain that receives synaptic inputs from striatum and sends projections to thalamus and/or brainstem nuclei. To define SNr in space, however, the only ways to delineate SNr clearly is to refer to TH staining that dissociates SNr from neighboring SNc or VTA, or use nuclear (or cell body) staining such as DAPI (or NeuN) to highlight the cyto-architecture. For this reason, we performed TH in Figure 3E, and DAPI staining for Figure 4 in the original experiments. Based on the reviewer’s suggestion, we have now intensified the DAPI’s brightness in Figure 4, and demonstrated DAPI staining in Figure 3E as well.

Admittedly, the anatomical definition of “medial” or “lateral” SNr is relative, based mainly on the spatial distribution of striatal projections. In our paper, we quantified SNr cells for the monosynaptic rabies tracing (Figure 3) in a principled way to define those by measuring distance between the medial and lateral edges of SNr (Figure 3—figure supplement 1), where the boundary was determined by the spatial references to TH staining and DAPI. In summary, we have now included better DAPI staining as a background in all images containing SNr (updated Figure 3E, Figure 4B, 4E, and 4G) as the reviewer suggested. Hopefully now the readers can see SNr clearly with local landmarks (such as the cerebral peduncle, cp) as references, to more clearly characterize the extent to which the observed axons/cell bodies are located in medial versus lateral SNr.